# *Parascedosporium putredinis* NO1 tailors its secretome for different lignocellulosic substrates

Conor J. R. Scott,[1] Nicholas G. S. McGregor,[2] Daniel R. Leadbeater,[1] Nicola C. Oates,[1] Janina Hoßbach,[1] Amira Abood,[1] Alexander Setchfield,[1] Adam Dowle,[3] Herman S. Overkleeft,[4] Gideon J. Davies,[2] Neil C. Bruce[1]

**ABSTRACT** *Parascedosporium putredinis* NO1 is a plant biomass-degrading ascomycete with a propensity to target the most recalcitrant components of lignocellulose. Here we applied proteomics and activity-based protein profiling (ABPP) to investigate the ability of *P. putredinis* NO1 to tailor its secretome for growth on different lignocellulosic substrates. Proteomic analysis of soluble and insoluble culture fractions following the growth of *P. putredinis* NO1 on six lignocellulosic substrates highlights the adaptability of the response of the *P. putredinis* NO1 secretome to different substrates. Differences in protein abundance profiles were maintained and observed across substrates after bioinformatic filtering of the data to remove intracellular protein contamination to identify the components of the secretome more accurately. These differences across substrates extended to carbohydrate-active enzymes (CAZymes) at both class and family levels. Investigation of abundant activities in the secretomes for each substrate revealed similar variation but also a high abundance of "unknown" proteins in all conditions investigated. Fluorescence-based and chemical proteomic ABPP of secreted cellulases, xylanases, and β-glucosidases applied to secretomes from multiple growth substrates for the first time confirmed highly adaptive time- and substrate-dependent glycoside hydrolase production by this fungus. *P. putredinis* NO1 is a promising new candidate for the identification of enzymes suited to the degradation of recalcitrant lignocellulosic feedstocks. The investigation of proteomes from the biomass bound and culture supernatant fractions provides a more complete picture of a fungal lignocellulose-degrading response. An in-depth understanding of this varied response will enhance efforts toward the development of tailored enzyme systems for use in biorefining.

**IMPORTANCE** The ability of the lignocellulose-degrading fungus *Parascedosporium putredinis* NO1 to tailor its secreted enzymes to different sources of plant biomass was revealed here. Through a combination of proteomic, bioinformatic, and fluorescent labeling techniques, remarkable variation was demonstrated in the secreted enzyme response for this ascomycete when grown on multiple lignocellulosic substrates. The maintenance of this variation over time when exploring hydrolytic polysaccharide-active enzymes through fluorescent labeling, suggests that this variation results from an actively tailored secretome response based on substrate. Understanding the tailored secretomes of wood-degrading fungi, especially from underexplored and poorly represented families, will be important for the development of effective substrate-tailored treatments for the conversion and valorization of lignocellulose.

**KEYWORDS** *Parascedosporium putredinis* NO1, lignocellulose, proteomics, CAZymes, activity-based protein profiling

Address correspondence to Conor J. R. Scott, cs1535@york.ac.uk, or Neil C. Bruce, neil.bruce@york.ac.uk.

The authors declare no conflict of interest.

See the funding table on p. 20.

Lignocellulosic residues from the waste components of established food crops (1–3) and from dedicated biomass crops (4) are a large and underutilized source

of biomass for biorefinery-based production of chemicals and biofuels. The major constituents of lignocellulose are cellulose, hemicelluloses, lignin, and minor amounts of pectins and nitrogen compounds (5). This intricate and insoluble network of polysaccharides and aromatic polymers makes lignocellulose a difficult substrate to degrade; however, a full understanding of the deconstruction of this complex substrate can offer a valuable source of sustainable fuels, chemicals, and materials for a wide range of applications (6–9).

Biological pre-treatments in the form of enzyme cocktails can deconstruct plant biomass as an environmentally friendly alternative to traditional physicochemical methods. Deployment of these enzymatic treatments has been limited by the hydrophobic and recalcitrant nature of lignocellulose. The heterogeneity of the substrate demands a diverse array of enzymes and treatment can be limited by the release of inhibitor compounds during biomass breakdown (10). This process is further complicated by the differing structures and compositions between lignocellulose sources, each ideally requiring a tailored enzyme cocktail, something that current commercial cocktails do not address (11).

Much research has focused on natural lignocellulose degrading microorganisms for enzymatic solutions to these issues (12, 13). Of particular significance, fungi are widely recognized as efficient degraders of plant biomass and major contributors to the carbon cycle of forest ecosystems. Across the fungal kingdom, a broad range of strategies are used in the deconstruction of lignocellulose. This includes the secretion of arrays of oxidative and hydrolytic enzymes to attack the polysaccharide components of plant cell walls, the production of high levels of lignin-degrading peroxidase and laccase enzymes to attack the lignin directly, and the generation of strong oxidants, via non-enzymatic radical based reactions to degrade lignocellulose indirectly (14, 15). Ascomycete soft-rot fungi are interesting sources of lignocellulose degrading enzymes because they often rely on a strategy of penetrating plant secondary cell walls where they can secrete large amounts of enzymes directly at the site of attack (16).

*Parascedosporium putredinis* NO1 is a soft-rot ascomycete identified from a mixed microbial community grown on wheat straw (17). *P. putredinis* NO1 dominated the fungal population in the later stages of the community culture, suggesting an ability to metabolize the more recalcitrant carbon sources in the substrate. This makes *P. putredinis* NO1 an interesting candidate for the identification of new lignocellulose-degrading enzymes. Indeed, a new oxidase enzyme was recently identified in the *P. putredinis* NO1 secretome which cleaves the major β-ether units in lignin to enhance the digestibility of wheat straw releasing tricin, coumaric acid, and vanillic acid in the process (17). Understanding the secretome response of a fungus that produces new lignocellulose-degrading enzymes, and which may target the recalcitrant components of lignocellulose, will be important for the development of effective methods to valorize lignocellulose.

An annotated reference genome for this fungus now provides a resource for a more effective investigation of the *P. putredinis* NO1 secretome (18). Proteomic analysis revealed an extensive and diverse array of carbohydrate-active enzymes (CAZymes) produced by NO1 during growth. The ability of *P. putredinis* NO1 to grow on various industrially relevant lignocellulosic substrates or kraft lignin has revealed that it is a versatile lignocellulose, particularly lignin, degrader.

To understand what genes are expressed during complex biomass degradation and to what degree these vary depending on the lignocellulosic substrate, the secretome during the growth of *P. putredinis* NO1 on six lignocellulosic substrates was investigated using both proteomic and activity-based protein profiling (ABPP) techniques. Proteins were harvested from both the supernatant and substrate-bound fractions of liquid cultures to give a more complete picture of the lignocellulose degrading proteome. The variability of this lignocellulose-degrading protein complement across substrates evidences a highly dynamic and tailored response to substrates by *P. putredinis* NO1. The enzymes identified here represent the vanguard of biomass-degrading capacity produced by a fungus that is highly effective at degrading lignin-enriched biomass.

Understanding this complex enzyme system will aid the design of feedstock-matched enzyme cocktails to enhance biorefining efficiency.

## RESULTS AND DISCUSSION

### Analysis of the *P. putredinis* NO1 secretome on different lignocellulosic substrates

To investigate whether the secretome of *P. putredinis* NO1 varies with lignocellulosic substrates, the fungus was grown on six substrates: oil palm empty fruit bunch, kraft lignin, rice straw, sugar cane bagasse, wheat bran, and wheat straw, and the resulting proteomes were sampled and analyzed. Proteins were harvested from culture supernatants and via a surface protein labeling approach with biotin that harvests proteins bound to substrates (19). It should be noted that kraft lignin is more soluble than the other substrates and therefore a larger portion of the kraft lignin bound fraction likely reflects proteins associated with the fungal cell wall. After harvesting and identifying proteins through label-free liquid chromatography–mass spectrometry (LC–MS), molar percentage values were generated for each protein. Quality control was performed by investigating protein count, performing principal component analysis, and clustering for all replicates. Through this analysis, a bound fraction sugar cane bagasse replicate, a supernatant fraction sugar cane bagasse replicate, a supernatant kraft lignin replicate, and a supernatant fraction wheat bran replicate were identified as outliers and removed from further analysis.

Across all samples 2,014 proteins were identified in at least one replicate, 1,890 proteins were identified among the substrate-bound fractions, and 973 were identified among the supernatant fractions, with 849 proteins shared across both fractions (Fig. S1). Many intracellular proteins, which were likely the results of cell death and lysis, were present in the proteomic data set despite efforts to target the extracellular fractions (20). The proteomic data were therefore filtered to remove the contaminating intracellular proteins according to the secretome isolation bioinformatics workflow presented previously (21). Secretome proteins were predicted to be extracellular using both BUSCA and DeepLoc localization prediction tools (20, 22), or were predicted to encode a secretion signal by SignalP, TargetP, and SecretomeP tools (23–25). The workflow utilizes both localization prediction and secretion signal prediction to capture both conventionally and unconventionally secreted fungal proteins (26). The vesicle-mediated release of lignocellulose degrading enzymes has been reported previously for *Trichoderma reesei*, another ascomycete degrader of plant biomass, highlighting the importance of capturing proteins secreted through unconventional pathways (27). Proteins that contained more than one predicted transmembrane domain by TMHMM or contained a single predicted transmembrane helix with more than 10 amino acids of this helix occurring in the first 60 amino acids of the protein sequence were then removed (28). The final secretome contained 228 proteins, a significant reduction from the total proteome data set. In the substrate-bound fraction of the secretome 165 proteins were identified in at least a single replicate, in the culture supernatant fraction 185 proteins were identified, and 122 proteins were shared between both fractions (Fig. 1). The now much smaller number of proteins identified exclusively in the bound and supernatant fractions were 43 and 63 proteins, respectively. Considering each substrate individually; for empty fruit bunch 107 proteins were identified in the bound fraction, 109 exclusively in the supernatant fraction, and 58 shared between both; for Kraft lignin, 85 proteins were identified in the bound fraction, 20 in the supernatant fraction, and 14 shared between both; for rice straw 130 proteins were identified in the bound fraction, 100 in the supernatant fraction, and 60 shared between both; for sugar cane bagasse 105 proteins were identified in the bound fraction, 104 in the supernatant fraction, and 66 shared between both; for wheat bran 72 proteins were identified in the bound fraction, 150 in the supernatant fraction, and 47 shared between both; wheat straw 101 proteins were identified in the bound fraction, 144 in the supernatant fraction, and 77 shared between both. The difference between the substrate-bound and culture

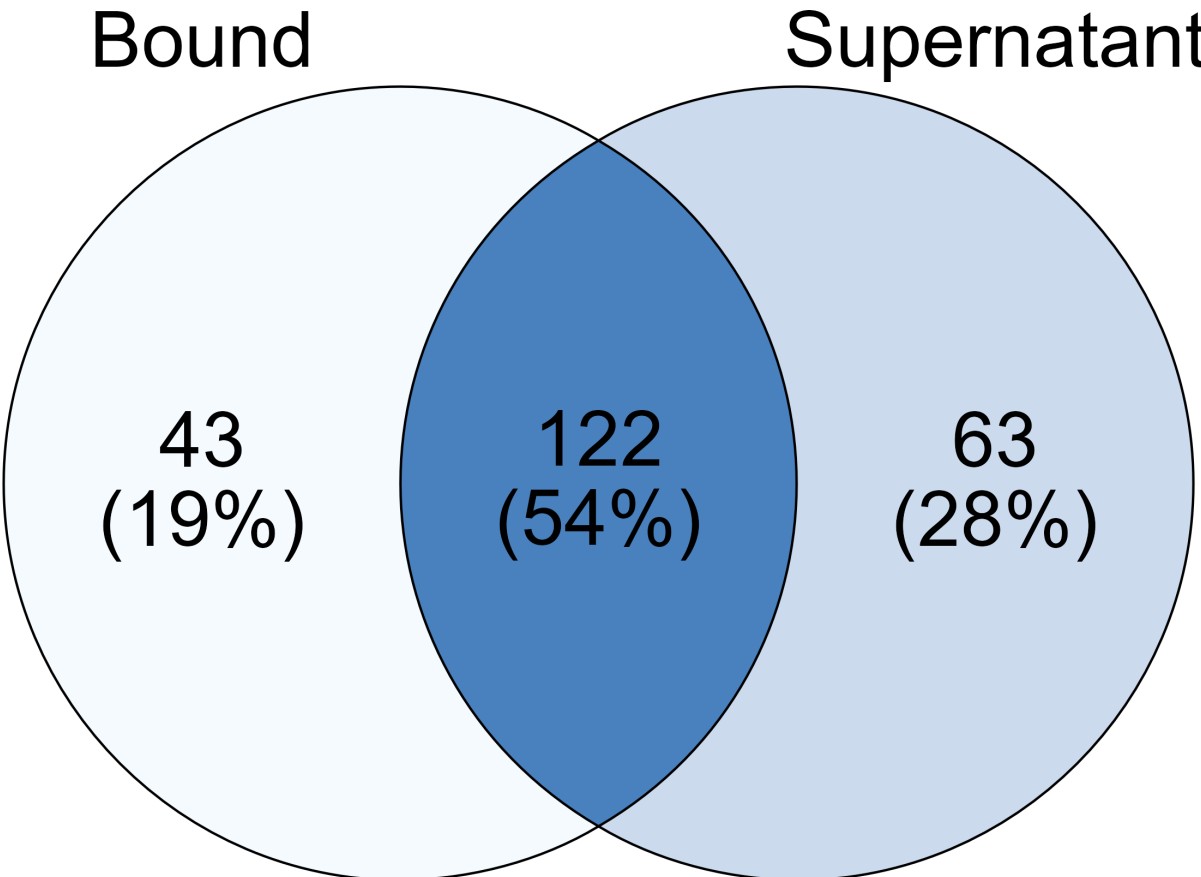

**FIG 1** Investigating the distribution of proteins across bound and supernatant fractions in the *P. putredinis* NO1 secretome. The number of proteins identified in at least one replicate across all substrates for the bound fraction compared to the supernatant fraction within the predicted *P. putredinis* NO1 secretome from growth on six lignocellulosic substrates for 4 days.

supernatant fractions highlights the importance of capturing both fractions, in order to fully understand the secretome response. The limitations of this workflow have been addressed previously (21), and it is accepted that some intracellular proteins may remain in the predicted secretome. However, the increased proportion of proteins annotated with extracellular functions achieved through this workflow demonstrates the removal of large numbers of intracellular proteins and therefore improves any interpretation of the *P. putredinis* NO1 secretome.

The overall protein abundance profiles of the *P. putredinis* NO1 secretome after growth on each substrate were visualized by scaling molar percentage abundance for each protein across substrates and clustering (Fig. 2). The substrate-dependent variation in the *P. putredinis* NO1 secretome was visible, with all substrates displaying substrate exclusive protein profiles to varying extents in both fractions. The differences between substrates were also investigated through principal component analysis, and clustering (Fig. S2 and S3). In the substrate-bound fraction, replicates grouped very well within substrates in principal component analysis (PCA) plots and clustered well within substrates in dendrograms, while demonstrating good separation between substrates in PCA plots and dendrograms also. In the supernatant fraction, the differences between substrates were less clear, with single replicates from some substrates grouping away from other replicates of the same substrate; however, this is likely due to the lower identified protein count for these replicates in the supernatant fraction. The lower number of replicates for the supernatant fractions of some substrates, and the lower protein counts in the supernatant fractions of some replicates reflect the difficulty in applying a single method of proteome harvest to cultures containing different substrates

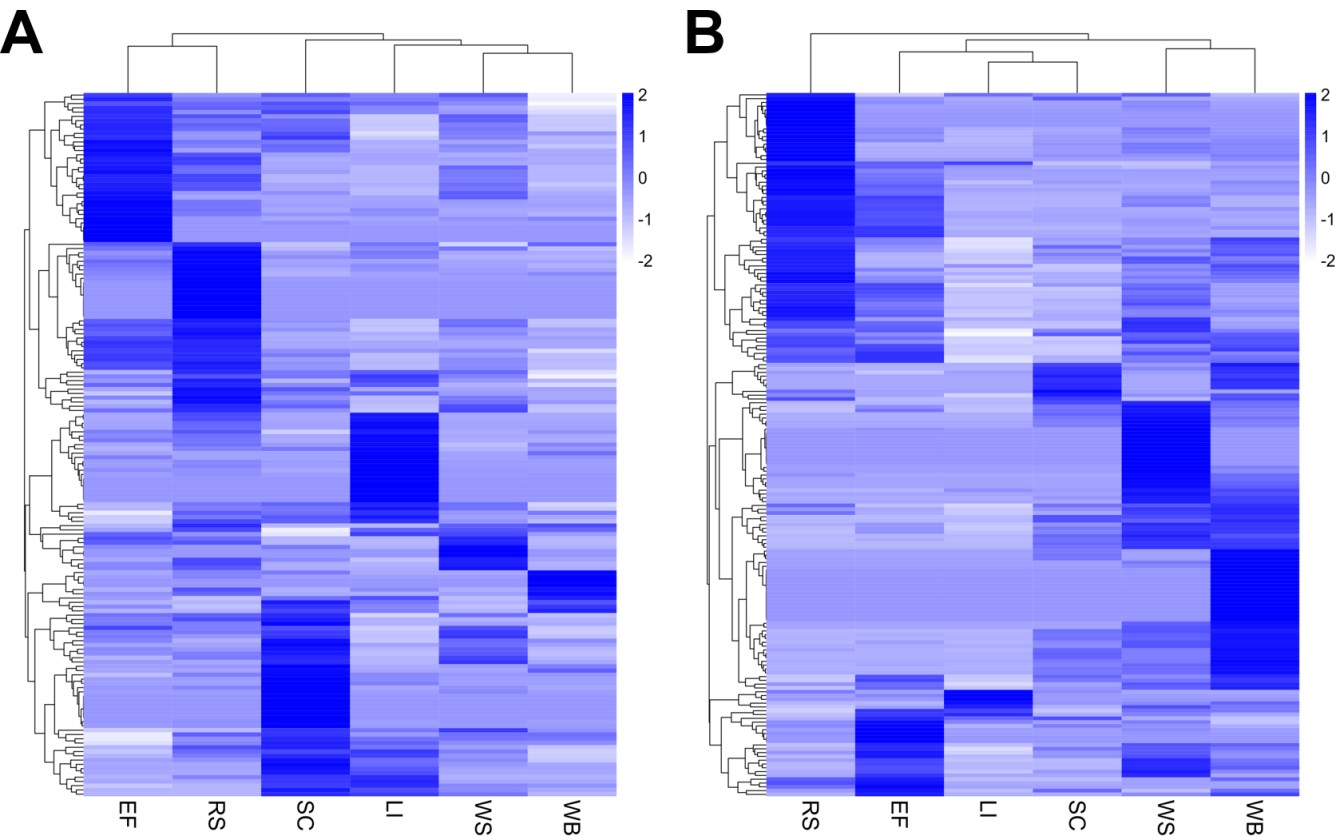

**FIG 2** Differences in the *P. putredinis* NO1 secretome across lignocellulosic substrates. Molar percentage values for proteins identified on at least one substrate in the bound (A) and supernatant (B) fractions of the *P. putredinis* NO1 secretome scaled to Z-scores across substrates. EF, empty fruit bunch; LI, kraft lignin; RS, rice straw; SC, sugar cane bagasse; WB, wheat bran; WS, wheat straw.

and therefore different conditions. This could also limit the ability to perform statistical comparisons between substrates. However, for this study, these replicates were still suitable for the visualization and comparison of the overall profiles of the *P. putredinis* NO1 secretome across substrates. The heatmaps in Fig. 2 still demonstrate that clear differences are visible across substrates in both fractions. This is especially interesting considering the somewhat similar proportions of cellulose, hemicellulose, and lignin for the empty fruit bunch from oil palm (29), rice straw (30), sugar cane bagasse (31), and wheat straw (32). Differences in fungal secretomes have been demonstrated previously when grown on compositionally distinct carbon sources (33–35). However, the variation on broadly similar lignocellulosic substrates demonstrated here is important as it may suggest a tailoring of the lignocellulose-degrading enzymes produced by *P. putredinis* NO1 which could be utilized for substrate-specific cocktail development. When *T. reesei* was grown on compositionally similar cellulose substrates it was suggested that secretome variation may be influenced by physical and structural differences (33), and the differences between the substrates investigated here may influence the variability of the *P. putredinis* NO1 secretome. Further study of these more subtle structural, compositional, and physical differences could improve the understanding of how fungal secretomes are influenced and tailored accordingly. Wheat bran has a different overall composition than the other lignocellulosic substrates with an increased starch content, correspondingly decreased proportions of cellulose and hemicellulose, and a reduced lignin content (36). The kraft lignin substrate is expected to contain minimal polysaccharides and instead is composed predominantly of branched polymeric lignin containing native lignin bonding patterns and new structures created during the kraft pulping

process (37). The difference in the secretome abundance profiles after the growth of *P. putredinis* NO1 on wheat bran and kraft lignin was therefore expected.

## An investigation of the functional profile of the *P. putredinis* NO1 secretome on different lignocellulosic substrates

To investigate the functional profiles of the *P. putredinis* NO1 secretome across lignocellulosic substrates, proteins were annotated with CAZyme domains and GO categories and terms. The overall functional profiles were investigated by comparing the proportional abundance of proteins annotated with CAZyme domains, if not annotated as CAZymes then assigned to GO categories, or not annotated with either (unknown) (Fig. 3). Clear variation in the proportions of functional category assignments were observed across substrates in both bound and supernatant fractions. The differences between bound and supernatant fractions for individual substrates validate further the importance of capturing both fractions when exploring a secretome in this context, as each fraction contains distinct profiles of enzyme activities. In the bound fraction (Fig. 3A), CAZymes contribute the most to total abundance for all substrates. This contribution is slightly reduced for the kraft lignin substrate, which is to be expected due to the low abundance of polysaccharides on which many CAZymes act (37). In the supernatant fraction (Fig. 3B), CAZymes again contribute the most to total abundance except for rice straw. Rice straw has a lower lignin content than some of the other substrates at 12% which may suggest increased accessibility to polysaccharide components and a reduced requirement for the production of oxidative lignin degraders (30). Structural aspects of lignocellulose such as cellulose structure and accessibility, pore size and distribution, and the extent and nature of lignin-carbohydrate complexes have all been reported to influence the efficiency of enzymatic hydrolysis (38). However, the relationship between such physical characteristics and microbial enzyme production is poorly understood. Surprisingly, CAZymes contributed the most to total abundance in the supernatant

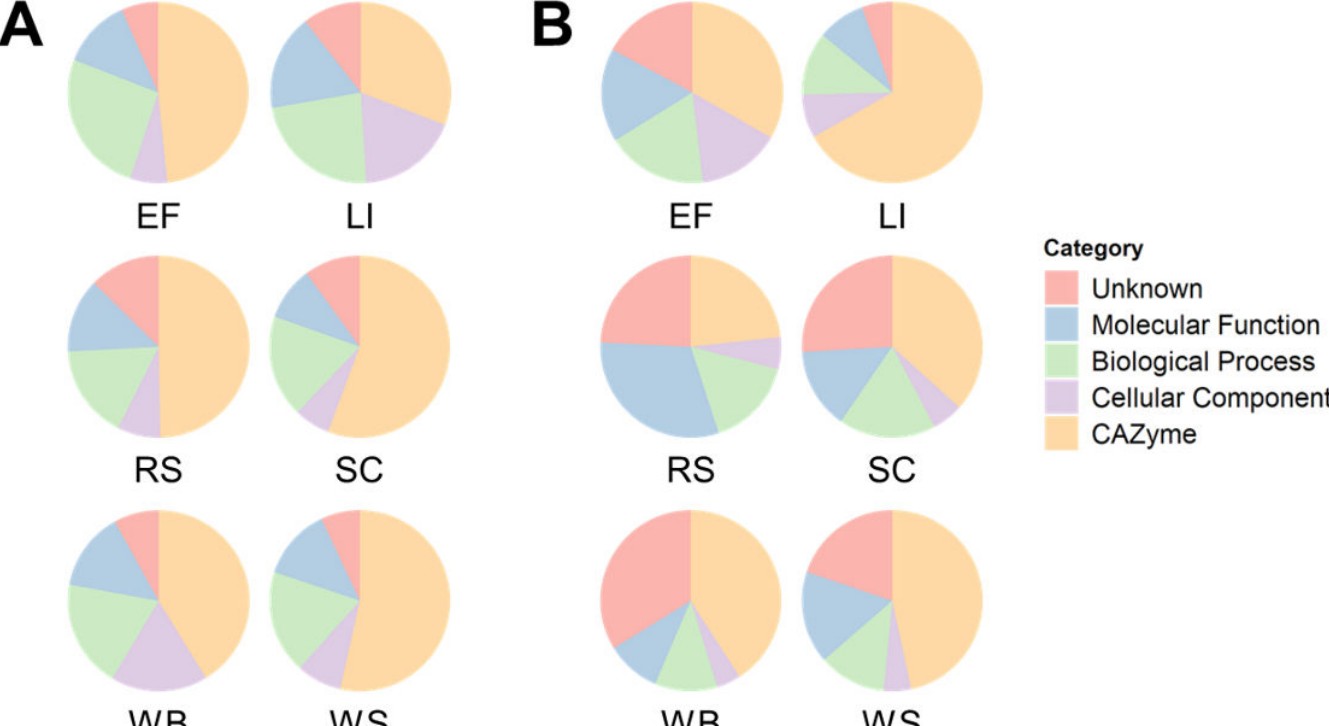

**FIG 3** Differences in functional categories of the *P. putredinis* NO1 secretome across lignocellulosic substrates. The molar percentage abundance of proteins identified as CAZymes, that were not identified as CAZymes but assigned to GO categories, or that were not annotated as CAZymes or with GO categories (unknown) were calculated proportionally for each substrate of the bound (A) and supernatant fractions (B) of the *P. putredinis* NO1 secretome.

fraction for kraft lignin which, although poorly soluble in water, lacks the polysaccharide chains targeted by binding modules (33, 39). Despite bioinformatic filtering of proteomic data to produce the *P. putredinis* NO1 secretome, some intracellular and membrane proteins may have persisted, as evidenced by the contribution of "cellular component" GO category proteins to abundance for all substrates in both bound and supernatant fractions. The persistence of proteins with predicted intracellular functions was reported previously in the development of the secretome isolation workflow applied here (21). The interference of these potentially intracellular proteins could have been reduced through stricter filtering when obtaining the *in silico* secretome; however, this could risk removing unknown proteins that are important for lignocellulose degradation. In both the bound and supernatant fractions, unknown proteins contribute similar proportions of abundance as the functional GO categories and again the proportion of these unknown proteins varies across substrates in both fractions investigated.

To understand the functional profiles of the *P. putredinis* NO1 secretome in more detail, the proportional abundance of the ten most abundant GO terms was plotted for each substrate for the bound and supernatant fractions (Fig. 4 and 5). For all substrates in both the bound (Fig. 4) and supernatant (Fig. 5) fractions, the top GO term annotations contributing to total abundance were predominantly extracellular activities associated with the hydrolytic breakdown of lignocellulose, agreeing with the polysaccharide abundance in many of the substrates (29–32, 36). In both fractions, many substrates included membrane annotations in the top ten most abundant GO terms. This may result from the identification of proteins in the workflow which may be targeted to the secretory pathway, but that is not necessarily secreted and instead fulfill roles in membranes (40). Another possibility is the binding and extraction of extracellular cell surface proteins by the biotin labeling technique, as biotinylation is a common technique for surface proteomic studies (41). In the bound fraction, membrane GO annotations contributed the highest abundance of any GO term for kraft lignin (Fig. 4B) and wheat bran (Fig. 4E), which may reflect the reduced CAZyme proportions seen previously for these substrates in this fraction. Once again, kraft lignin and wheat bran, the two most compositionally distinct substrates explored here demonstrate the most distinct secretome activity profiles. Aside from cell surface proteins, abundant intracellular annotations were largely absent from most of the secretomes except for the kraft lignin substrate. Unknown proteins, i.e., with no GO annotations, were also present in the top ten most abundant annotations for all substrates in both fractions except for the supernatant fraction of kraft lignin. In the supernatant fraction, unknown proteins made the largest contribution of any annotation for empty fruit bunch (Fig. 5A), rice straw (Fig. 5C), sugar cane bagasse (Fig. 5D), wheat bran (Fig. 5E), and wheat straw (Fig. 5F). The most abundant annotations for most substrates in both fractions appear to be associated with the breakdown of cellulose and xylan, two abundant polysaccharides in these lignocellulosic substrates (29–32, 36). However, there is a clear difference in profile for the kraft lignin substrate in both the bound (Fig. 4B) and supernatant (Fig. 5B) fractions. In the bound fraction, there appear to be more membrane-targeted GO annotations in the ten most abundant annotations. Finally, the GO annotation profile of the bound fraction for wheat bran (Fig. 4E) distinctly contained chitin binding and catabolizing annotations. Wheat bran has been investigated for its use as a substrate for the induction of chitinase production in fungal species previously and perhaps a similar induction is occurring for *P. putredinis* NO1 (42).

## The lignocellulose degrading secretome of *P. putredinis* NO1 varies depending on the growth substrate

To investigate the lignocellulose-degrading enzyme repertoire of the *P. putredinis* NO1 secretome, protein sequences were annotated for CAZyme domains using the dbCAN server (43). CAZyme proteins are assigned to the following catalytic classes: auxiliary activity (AA), carbohydrate esterase (CE), glycoside hydrolase (GH), glycosyl transferase

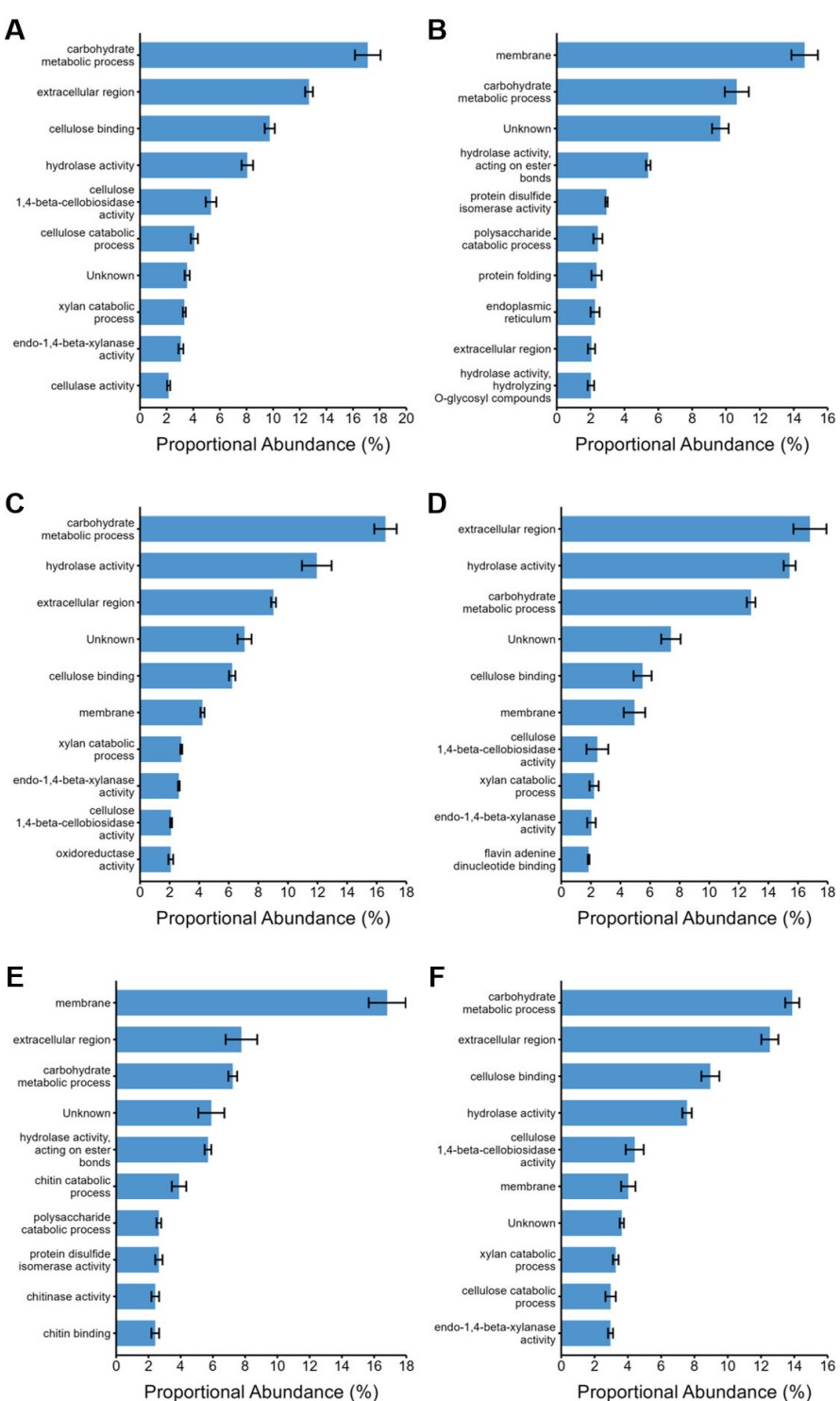

**FIG 4** Proportional abundances of enzyme activities in the bound fraction of the *P. putredinis* NO1 secretome. The molar percentage abundance of proteins assigned to GO terms was calculated proportionally for the bound fraction of the *P. putredinis* NO1 secretome after 4 days of growth on six lignocellulosic substrates. EF, empty fruit bunch (A); LI, kraft lignin (B); RS, rice straw (C); SC, sugar cane bagasse (D); WB, wheat bran (E); WS, wheat straw (F). Average molar percentage ± standard error ($n = 3$, $n = 2$ for SC substrate).

(GT), and polysaccharide lyase (PL). Additionally, many CAZymes have carbohydrate-binding modules (CBMs) which are also assigned.

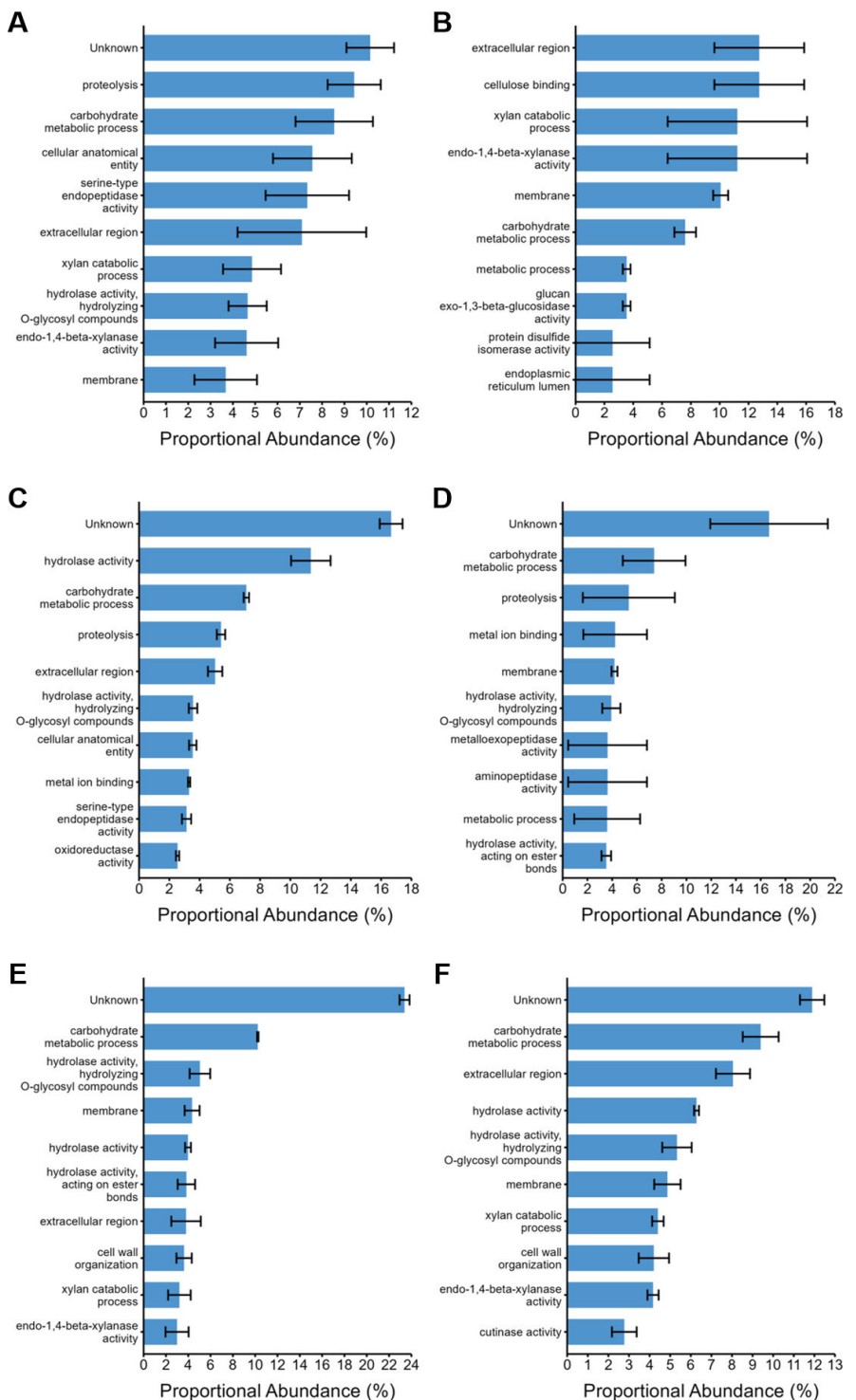

**FIG 5** Proportional abundances of enzyme activities in the supernatant fraction of the *P. putredinis* NO1 secretome. The molar percentage abundance of proteins assigned to GO terms was calculated proportionally for the supernatant fraction of the *P. putredinis* NO1 secretome after 4 days of growth on six lignocellulosic substrates. EF, empty fruit bunch (A); LI, kraft lignin (B); RS, rice straw (C); SC, sugar cane bagasse (D); WB, wheat bran (E); WS, wheat straw (F). Average molar percentage ± standard error ($n = 3$, $n = 2$ for SC, LI, and WB substrates).

Comparing the contribution of proteins assigned to these catalytic classes to total CAZyme abundance across substrates demonstrates similar variation seen for the total

secretome and for functional categories in both bound and supernatant fractions (Fig. 6). In the bound fraction AA class CAZymes contribute a high proportion of total CAZyme abundance for the empty fruit bunch, rice straw, sugar cane bagasse, and wheat straw substrate but are low in abundance for wheat bran and kraft lignin substrates (Fig. 6A). Crystalline cellulose is present in lower levels or is absent from wheat bran and kraft lignin substrates, and this may reflect a reduced production of AA class CAZymes involved in lytic polysaccharide monooxygenase (LPMO) systems which degrade crystalline cellulose oxidatively (36, 37, 44, 45). Potentially, the oxidative enzymes responsible for lignin breakdown are still present but in low abundance. The reductions in AA abundances for wheat bran and kraft lignin substrates correspond with increased proportions of all other CAZyme classes in the bound fraction, including GT class CAZymes (46). Some GT enzymes are extracellular and bound to the fungal cell surface which is likely why these proteins have been isolated in the *P. putredinis* NO1 secretome and they are more likely to play a role in fungal growth, cell wall remodeling, or potentially associate the fungus with the growth substrate (47, 48). PL class enzymes were absent from both fractions of the *P. putredinis* NO1 secretome during growth on kraft lignin. They were either absent or present in low abundances for all substrates in the bound fraction but were present with varying contributions to total CAZyme abundance in the supernatant fraction. This could be explained by the soluble pectin substrates on which PL enzymes act which are more likely to be present in the culture supernatant (49). In the supernatant fraction, GH class CAZymes dominated the CAZyme abundance profiles for all substrates (Fig. 6B). It can be hypothesized that soluble components of the substrates are present initially and are also released into

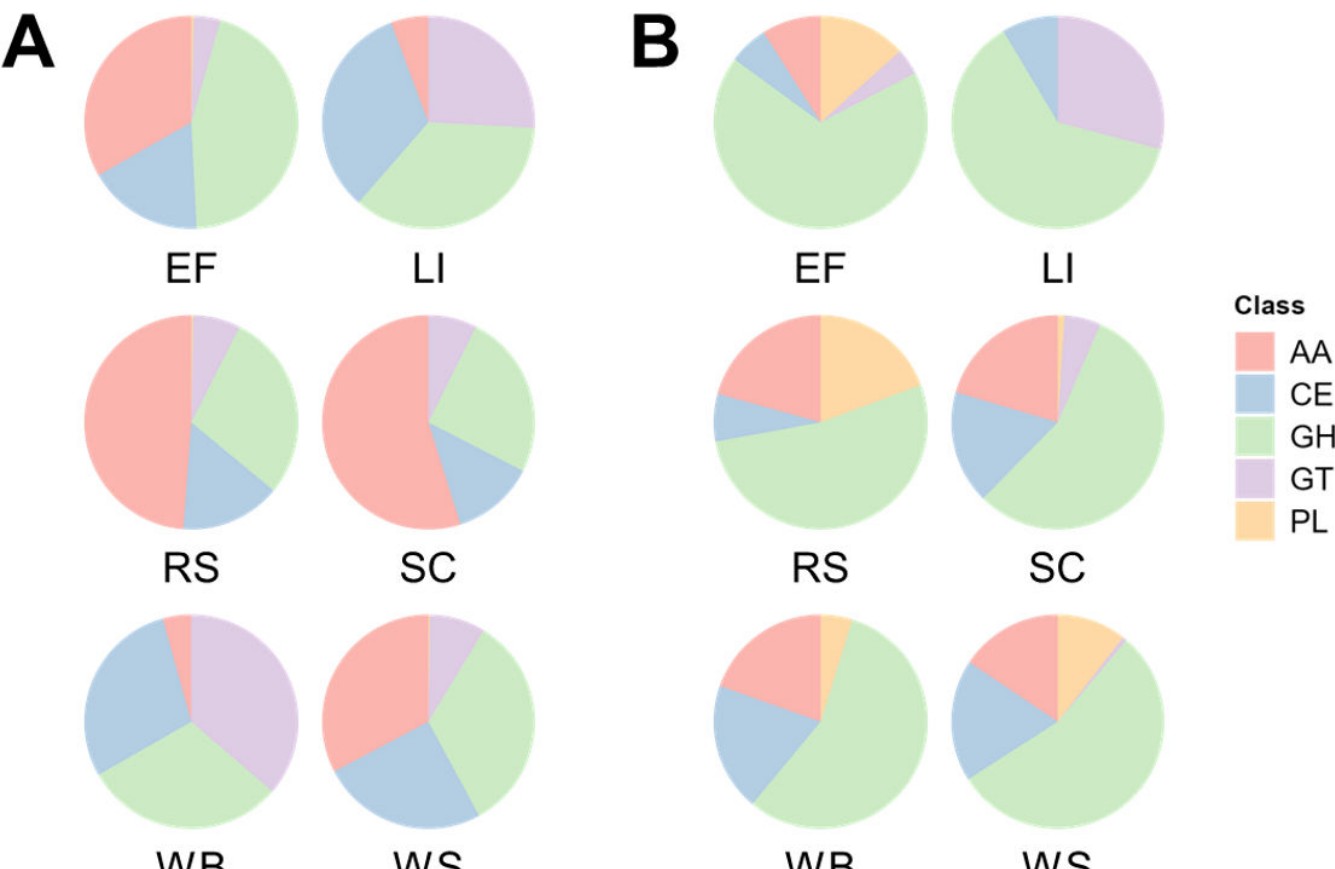

**FIG 6** Differences in proportional catalytic CAZyme class abundance of the *P. putredinis* NO1 secretome across lignocellulosic substrates. The molar percentage abundance of proteins belonging to each catalytic class of CAZyme was calculated proportionally to the total abundance of CAZymes for each substrate for the bound (A) and supernatant fractions (B) of the *P. putredinis* NO1 secretome.

the supernatant fraction during degradation. These soluble components would then be accessible to the wide array of hydrolytic GH class CAZyme activities which act on diverse substrates, and this potentially explains the abundance of these enzymes in this fraction (50). The similarity in the proportion of GH class CAZymes in the supernatant fraction even for compositionally distinct substrates such as wheat bran and kraft lignin could be the result of the transcriptional activation of GH class CAZymes like cellulases and xylanases. Mono- and disaccharides, which could be present in all substrates investigated here, have been demonstrated to act as inducers of the expression of large numbers of hydrolytic CAZymes in other ascomycete fungi previously (35, 51). This may be occurring for all substrates despite the target polysaccharides of induced enzymes not necessarily being present in similar amounts across all substrates investigated.

To investigate the CAZyme classes in more detail and to address some of the questions raised by comparing class profiles, the CAZyme family abundances were compared across substrates for the bound and supernatant fraction separately (Fig. 7). Variation across substrates was again clear at the CAZyme family level in both fractions even for substrates which looked similar at the class level. This highlights how similarities at the class level masked these underlying differences at the level of the CAZyme family, which defines an enzyme's activity more specifically. As expected, the relative abundances of AA9 LPMOs and other AA families, for example, AA3, AA7, and AA12 which act in LPMO systems are reduced for kraft lignin and wheat bran substrates in the bound fraction (52–54). However, the AA1 laccase family enzymes responsible for the oxidative cleavage of lignin structures are present in increased abundance in the bound fraction for the kraft lignin substrate (53). All AA families were muted in abundance in the bound fraction for wheat bran except for the AA13 family of starch-degrading LPMOs, likely resulting from the increased starch content of the substrate (36, 55). Variation in the abundance profiles of proteins with CBMs belonging to different families can also be seen in (Fig. 7). CBMs are non-catalytic domains that predominantly bind to plant cell wall polymers to prolong contact of catalytic CAZyme domains with their substrates to enhance efficiency (56). Harvesting the bound fraction of lignocellulose-degrading secretomes was performed in an attempt to capture proteins with such binding capacity, which may have been missed by harvesting proteins from the soluble supernatant fraction only. A wide range of CBM families provide binding capabilities for different enzymes to a range of cell wall components, and the variation in CBM containing protein abundance observed here may reflect the varying compositions and structures of the substrates (33). Indeed some expected patterns can be seen in the bound fraction, such as the relatively lower abundance of CBM1 proteins, involved in crystalline cellulose binding, to kraft lignin and wheat bran substrates which have reduced proportions of cellulose in comparison to the other substrates investigated (36, 37, 57). For wheat bran, in the bound fraction there is a relatively high abundance of CBM43, CBM5, and CBM56-containing proteins. CBM5 proteins are involved in binding chitin and are likely to be involved in cell-wall remodeling here (58). CBM43 and CBM56 are involved in binding β-1,3-glucans, which are present in fungal cell walls, but also present in the wheat bran substrate and are potentially being targeted for breakdown here (36, 59, 60). The hydrolytic families of CAZymes belonging to GH, CE, and PL classes show variation in abundance profiles across all substrates in both fractions and likely reflect a tailored hydrolytic response to the different polysaccharide compositions of each substrate. Although the cellobiose and xylose-mediated induction of hydrolytic CAZymes is documented for more established ascomycete fungi (35, 51, 61), whether these mechanisms exist in *P. putredinis* NO1 is not yet known. The variation in other classes of CAZyme in ascomycetes, where regulation has not been well investigated in the context of lignocellulose breakdown, also suggests additional mechanisms of activation of gene expression.

Generally, the abundances of the hydrolytic CAZymes are lower for the kraft lignin substrate in both fractions, agreeing with the reduced levels of polysaccharides. In the supernatant fraction, the relatively high abundance of some polysaccharide hydrolyzing

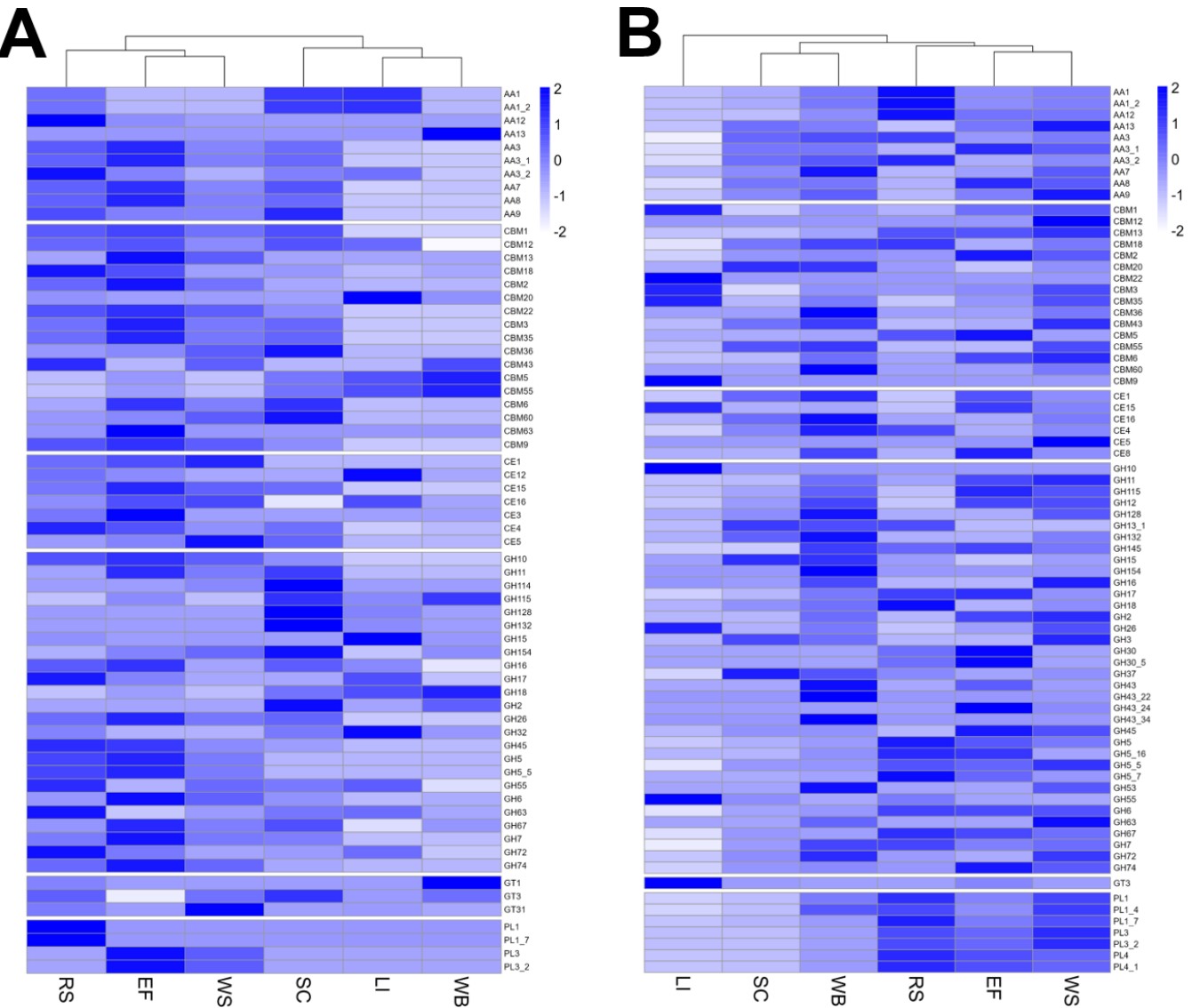

**FIG 7** Differences in CAZyme family abundance of the *P. putredinis* NO1 secretome across lignocellulosic substrates. Molar percentage values for proteins annotated as CAZymes and identified on at least one substrate of the *P. putredinis* NO1 secretome scaled to Z-scores across substrates for the bound (A) and supernatant (B) fractions separately.

or binding CAZyme family domains during growth on kraft lignin may reflect enzymes produced irrespective of the lignocellulosic growth substrate. Indeed, it has been observed in *T. reesei* that spore germination alone can lead to substantial upregulation of polysaccharide-degrading CAZymes to prepare the fungus for a habitat containing plant biomass (62). The GH10, GH26, GH55, and CE15 families produced in relatively high abundance for kraft lignin possibly represent a similar response. It is possible that the action of these enzymes, which may be produced irrespective of the substrate, releases oligo- or mono-saccharides that could lead to induction and tailoring of the secretome (35, 51, 61). Investigating CAZyme family profiles in the supernatant fraction also demonstrates reduced relative abundances for all CAZyme domain-containing proteins for the sugar cane bagasse substrate, and variation across wheat straw, rice straw, and empty fruit bunches from oil palm, despite all these substrates having similar overall biomass compositions (63, 64). This variation demonstrates the complexity of the fungal plant biomass-degrading response to different substrates which is still poorly understood.

The differences in CAZyme family abundance all suggest a potential ability of *P. putredinis* NO1 to tailor its secretome dependent on its lignocellulosic growth substrate. Empty fruit bunches from oil palm, rice straw, sugar cane bagasse, and wheat straw all show clear differences between the substrates as well, although the overall compositions are more similar (29–32). One limitation of the current study was that it was possible for the observed differences in secretome profiles to be a product of different fungal growth stages on each substrate, as all data were harvested at a single time point. An alternative approach was therefore taken to confirm that differences in the *P. putredinis* NO1 secretome are maintained over time.

## Activity-based protein profiling shows how lignocellulose-degrading enzyme profiles vary over time

Fluorescence-based ABPP with cyclophellitol-derived activity-based probes (ABPs) for retaining β-glucosidases (65), cellulases (66), and xylanases (67) have been used to screen fungal secretomes previously (67, 68). Cellulases, xylanases, and β-glucosidases are broadly distributed and highly expressed in plant-biomass degrading fungi, making them good candidates for ABPP techniques and for comparison across lignocellulolytic systems (68). These probes were also used to display the induction of an array of GH enzymes in basidiomycete secretome when grown on lignocellulosic compared to simple substrates like maltose (68). Therefore, this is an appropriate and effective technique to explore the potentially tailored secretome of the ascomycete *P. putredinis* NO1 during growth on lignocellulosic substrates.

Activity-based probes for GHs were employed to screen samples of supernatants harvested from cultures of *P. putredinis* NO1 grown on the same substrates as the proteomics experiment at days 3, 4, 5, 6, 7, and 10 (69). The biotin-labeling approach used in the proteomic experiments to sample bound fractions involves protein denaturation and is therefore incompatible with this ABPP technique which requires active site binding. The supernatants were used to screen for and to determine the relative levels of lignocellulose-degrading glycoside hydrolases. The activities were present in the empty fruit bunch from oil palm, rice straw, sugar cane bagasse, wheat bran, and wheat straw samples (Fig. 8). Unfortunately, the high aromatic content of the kraft lignin interfered with staining and meant that this substrate was incompatible with this technique (Fig. S4). Fluorescent gel visualization was performed for triplicate samples from all substrates and good agreement in hydrolase abundance pattern over time was observed within substrates (Fig. S4). For simplicity, single replicate time course fluorescence gels are presented in Fig. 8, and integrated band intensity values are resolved in Fig. 9.

As seen previously for screening of basidiomycete secretomes, the *P. putredinis* NO1 secretome shows differential GH production over time dependent on growth substrate (Fig. 8 and 9) (68). *P. putredinis* NO1 demonstrated muted production of GHs at day 3 for all substrates. For rice straw, this production remained low for the entire time course with low levels of xylanase and β-glucosidase and only a single cellulase detected at low abundances. Although the production profiles of these GHs were demonstrated to vary by species, it is worth noting that transcriptomic analysis of another ascomycete, *Thielavia terrestris,* only found a single cellulase gene to be in the most highly expressed genes when grown on rice straw compared to eleven cellulases for growth on Avicel (68, 70). It was also observed that compared to growth on glucose, predominantly oxidative AA LPMO family CAZymes and hemicellulose-active enzymes were up-regulated during *T. terrestris* growth on rice straw (70). The reason for the lack of cellulose targeting GHs also observed here is unclear, especially considering the reported induction of cellulases from both cellobiose and xylose in other ascomycetes (35, 51, 61), but may suggest a predominantly oxidative approach to rice straw deconstruction instead. Perhaps the distinct fluorescent profile observed here for rice straw may result from the unique structural properties of rice straw, such as its high silica content which may prevent access to polysaccharides (71). Relatively high abundances of AA3 and

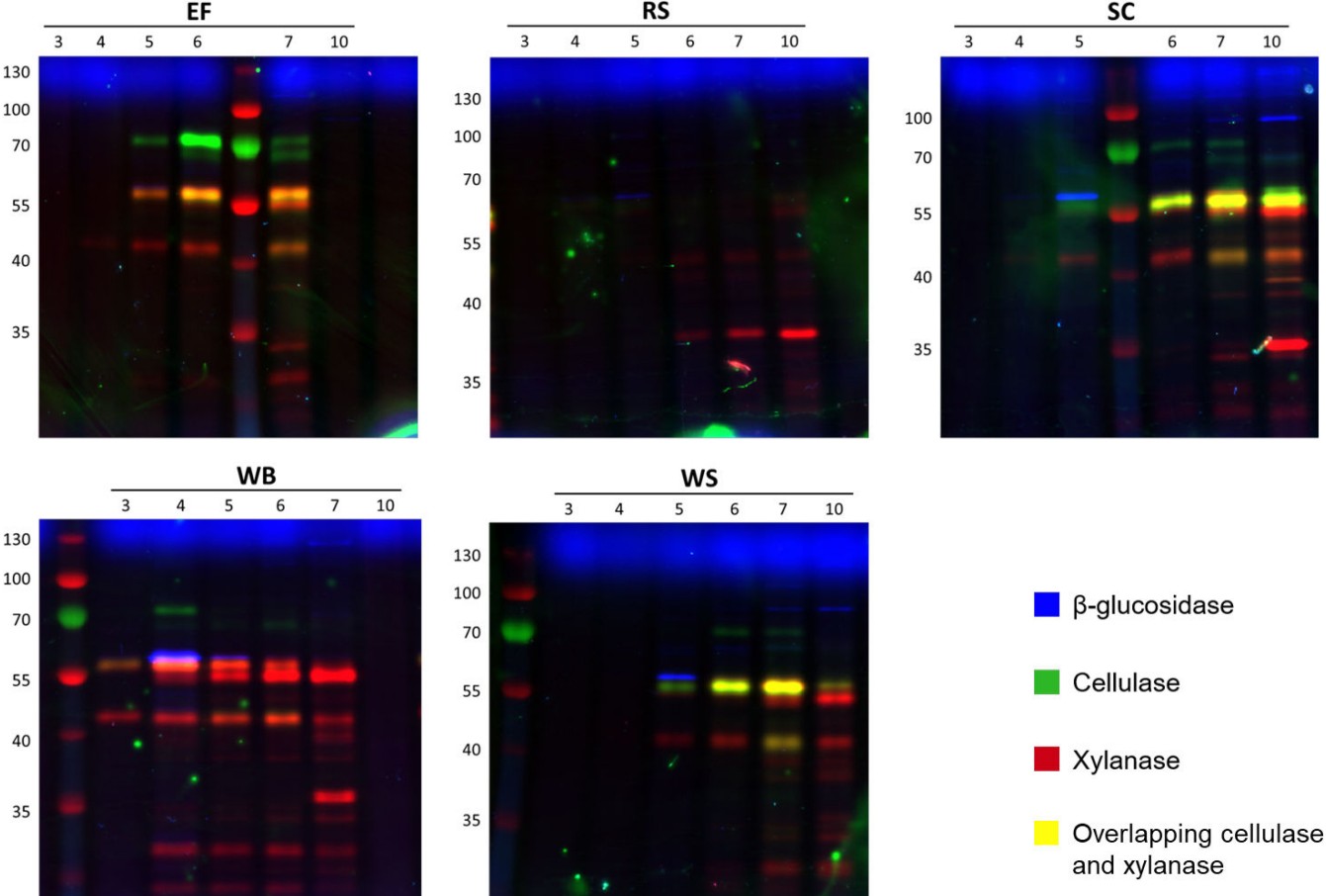

**FIG 8** Differences in *P. putredinis* NO1 glycoside hydrolase production over time visualized with activity-based probes. Fluorescence imaging following SDS-PAGE is shown for single replicates of samples of culture supernatants taken at days 3, 4, 5, 6, 7, and 10 of growth. They were treated with a triplex probe mixture targeting cellulases, xylanases, and retaining β-glucosidases. EF, RS, SC, WB, WS, alongside PageRuler (Thermo) prestained protein ladder. Secretomes were stained in triplicate, a single replicate is shown here.

AA12 family CAZymes were observed in the rice straw secretome supernatant fraction, where the probes are deployed (Fig. 7). These oxidative enzymes act within LPMO systems; however, relatively low abundances of the AA9 LPMOs were also observed. Further investigation specifically into growth on rice straw would be required to fully understand the strategy of lignocellulose breakdown adopted here by *P. putredinis* NO1 but is warranted considering the abundance and environmental issues posed by the vast amounts of rice straw generated annually and burnt on the field (72, 73).

Substrate-specific xylanase production patterns have been observed previously using the same fluorescent probe to investigate the growth of *Aspergillus niger* on beechwood xylan (67). The xylanase probe detected similar variation across substrates here, most noticeably for wheat bran where xylanase production was the most dominant of the three types of GH screened (Fig. 8). Cellulase production was detected despite being obscured on the gel images by the high xylanase signals (Fig. 9). However, cellulase production was not sustained until day 10 as it was for wheat straw, sugar cane bagasse, and empty fruit bunch. This reflects the pattern of reduced cellulose degrading CAZyme family abundance observed for wheat bran in the proteomic investigation, a substrate with reduced cellulose content (36). Although wheat bran has been explored as a potential substrate for cellulase production from the ascomycete *T. reesei,* where the nitrogen-rich substrate induces cellulase production (74, 75), this study demonstrates how varied fungal lignocellulose-degrading enzyme responses can be for the

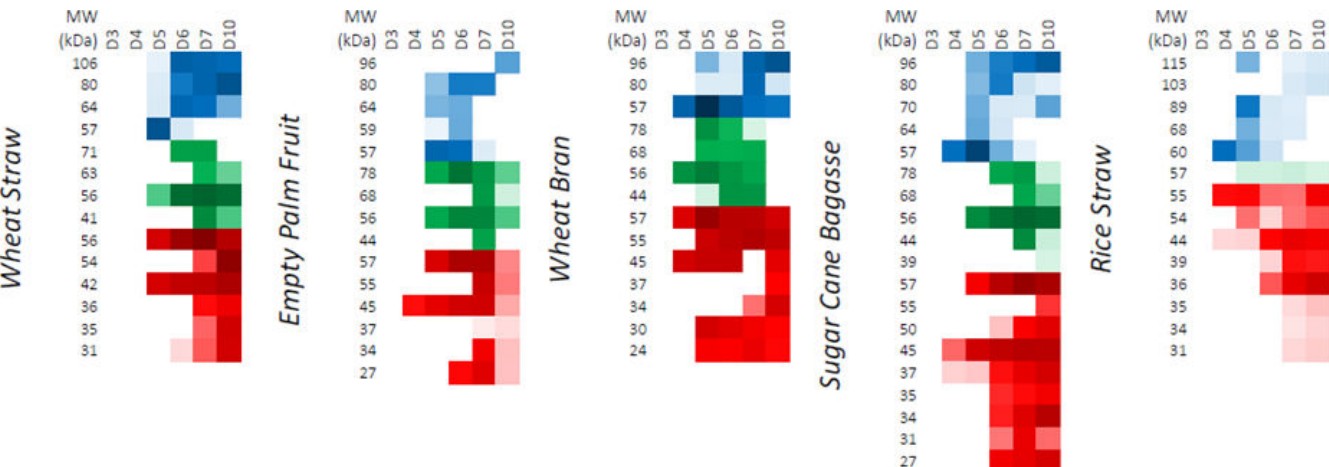

**FIG 9** ABPP-determined variation in relative active enzyme levels over time during *P. putredinis* NO1 growth on various substrates. Resolved bands running at different apparent MW values (left column in each block) were integrated into the Cy2 (β-glucosidase probe), Cy3 (cellulase probe), or Cy5 (xylanase probe) channels. Average band integration values (*n* = 3) are shown as color intensity varying from white (not detected) to full color (~1,000,000 counts) to black (saturation) on a logarithmic scale. Secretomes were prepared and stained in biological triplicate following different culture times (labels above columns).

same substrate and how characterization of new lignocellulose-degrading systems are important for understanding different approaches which may adopted industrially for the conversion of biomass.

GH CAZyme production patterns looked similar for wheat straw and sugar cane bagasse substrates aside from some additional low molecular weight xylanases detected for the latter. All substrates except rice straw showed the presence of a 57 kDa glucosidase at high levels on day 5 which then became undetectable by day 7 for wheat straw, empty fruit bunch, and sugar cane bagasse, often giving way to a series of higher MW glucosidases. This two-stage pattern of glucosidase production may represent distinct responses to water-soluble glucosides vs polysaccharide-derived glucosides. It has also been observed previously in microbial communities that β-glucosidase patterns can shift depending on other available carbon sources (76). Perhaps, the two-stage pattern of β-glucosidase production observed here also reflects a change in available carbon after day 5 of growth on these substrates.

Time-dependent analysis of the oil palm empty fruit bunch secretomes showed that induction was slow, with no enzymes detected at day 4. Day 5 showed a strong induction across all samples of the production of cellulases, glucosidases, and xylanases. Especially obvious was the sustained high production of a higher molecular weight cellulase detected across all substrates investigated (Fig. 8 and 9). The production of hemicellulose-degrading enzymes by *A. niger* on this substrate has been explored previously; however, exploration of cellulase production has not been performed for fungal isolates (77). The growth of the bacterium *Paenibacillus macerans* on empty fruit bunches from oil palm has been explored where degradation of the substrate was demonstrated (78). However, cellulase activity from this microorganism was not observed. Overall, this analysis demonstrates the time-dependent adaptability of the *P. putredinis* NO1 secretome to different lignocellulosic substrates and illuminates the different CAZyme specificities deployed by this fungus. Considering the potential lignin-degrading lifestyle of *P. putredinis* NO1, from which a new lignin-degrading oxidase has been identified previously (17), the investigation of oxidative enzymes in the secretome and their variation dependent on substrate and over time would be of great interest. However, at present, the use of ABPP techniques to probe fungal secretomes as was performed here is limited to hydrolytic enzymes.

## Conclusions

Proteomic analysis of the secretome of *P. putredinis* NO1 grown on multiple lignocellulosic substrates showed a substrate-dependent variation of the CAZyme complement both bound to the biomass and in the culture supernatant. This variation was maintained when comparing the abundances of CAZymes at both the class and family levels. Patterns of abundance of important lignocellulose degrading enzymes were observed, for example, the lack of crystalline cellulose targeting enzymes during growth on substrates with reduced cellulose contents. When investigating CAZymes at the family level, which more specifically defines enzyme activity, the abundance profiles were found to vary greatly across substrates in both fractions of the lignocellulose degrading secretomes. This likely reflects the varying structures and compositions of the lignocellulosic substrates and suggests a potential ability for *P. putredinis* NO1 to actively tailor its enzymatic response.

It could be argued that differences in proteomic data across substrates reflected different stages of fungal growth on the different substrates. However, by utilizing an ABPP-based approach to visualize secretomes it was demonstrated that the diversity and titers of active cellulases, xylanases, and β-glucosidases in the secretomes varied both with substrate and over time. Therefore, the fungal growth stage is not solely responsible for differences observed between the secretomes. Patterns of production of these GH class CAZymes were also found which both agreed and contradicted previous reports on fungal GH expression on lignocellulosic substrates (35, 51, 61). This highlights how varied and complex lignocellulose degrading systems can be and demonstrates the value of exploring new systems like *P. putredinis* NO1 to better understand lignocellulose breakdown and to identify novel enzymes.

The complexity of fungal lignocellulose-degrading responses revealed here highlights the shortcoming of generic commercial cocktails for application to different substrates, but also the difficulty in disentangling enzyme effects and designing optimally tailored cocktails. Understanding the adaptability of fungal secretomes will allow increased efficiency in the depolymerization and biorefining of lignocellulosic substrates. Furthermore, the significant number of unknown proteins identified in the secretome after filtering through the bioinformatics pipeline suggests that there are still new lignocellulosic activities to be discovered.

## MATERIALS AND METHODS

### Strain isolation

*P. putredinis* NO1 was isolated from a wheat straw enrichment culture and maintained as reported previously (17). A mixed microbial community grown on wheat straw was characterized by 16S ribosomal and ITS region amplicon sequencing over 8 weeks of growth. ITS reads assigned to the genus *Graphium* dominated the eukaryotic community after 4 weeks of incubation. *P. putredinis* NO1, a synamorphy of *Graphium,* was readily cultivated from this community on potato dextrose agar plates.

### *P. putredinis* NO1 cultures for proteomics

Triplicate 500 mL solutions of media containing 1.5% (wt/vol) rice straw, wheat straw, sugarcane bagasse, wheat bran, empty fruit bunches from oil palm or containing 5% (wt/vol) kraft lignin were inoculated to a final concentration of $10^5$ spores/mL of *P. putredinis* NO1. Cultures were incubated at 30°C at 150 rpm for 4 days before harvesting for proteomic investigation. The optimized media contained KCl 0.52 g/L, $KH_2PO_4$ 0.815 g/L, $K_2HPO_4$ 1.045 g/L, $MgSO_4$ 1.35 g/L, $NaNO_3$ 1.75 g/L, yeast extract 8.85 g/L, and Hutner's trace elements and was based on *A. niger* media (79).

## Harvesting the *P. putredinis* NO1 proteomes across substrates

To harvest supernatant proteins, culture supernatants were centrifuged at 12,000 × *g* for 20 min at 4°C and were then sterilized through 0.22 µm PES filter units. Triplicate 5 mL technical replicates for each culture were combined with 5 volumes of ice-cold 100% acetone and mixed by inverting before incubation overnight at −20°C. Samples were then centrifuged at 4,500 rpm for 20 min at 4°C and the acetone supernatant discarded. The pellets were then washed twice by the addition of ice-cold 80% acetone, vortexing, and centrifuging at 4,500 rpm for 10 min at 4°C. Pellets were air dried and resuspended in 1 mL of 0.5× PBS buffer before transferring to Eppendorfs and snap freezing in liquid nitrogen and storing at −80°C.

To extract proteins bound to substrates, triplicate samples of 2 g of biomass from each culture were washed twice through the addition of 25 mL ice-cold 0.5× PBS and centrifugation at 4,500 rpm for 5 min at 4°C. The supernatant was discarded, and pellets were resuspended in 19 mL of 0.5× PBS with 1 mL of biotin (EZ-link-Sulfo-NHS-SS-biotin, Thermo Scientific) with rotation at 4°C for 1 h. Samples were then centrifuged at 4,500 rpm for 10 min at 4°C and the supernatant discarded. The reaction was quenched by the addition of 25 mL of 50 mM Tris-HCl pH 8.0 with rotation at 4°C for 30 min. The biomass was pelleted as previously, and the supernatants were discarded. Biomass pellets were washed twice with 20 mL 0.5× PBS and were resuspended in 10 mL of 2% SDS pre-heated to 60°C with rotation at 20°C for 1 h. Samples were centrifuged at 4,500 rpm for 10 min and the supernatant was collected. Five volumes of ice-cold 100% acetone were added, and samples were incubated at −20°C overnight. Samples were pelleted and washed in the same way as before for the supernatant fraction but were resuspended after air drying in 1 mL of 0.1% SDS solution before filtering through 0.22 µm PES filters.

Each replicate was loaded onto its own individual HiTrap Streptavidin HP 1 mL column (GE Healthcare) at a flow rate of 0.5 mL/min. Columns were left to incubate for 1 h at 4°C and were then washed with 10 mL of 0.1% SDS in PBS solution at a flow rate of 1 mL/min. Proteins were eluted by loading 1 mL of 50 mM DTT in PBS solution and incubating columns overnight at 4°C. Another 1 mL of of 50 mM DTT/PBS was added, and 1 mL of protein was eluted, the column was incubated for 1 h at 4°C, and then another 1 mL of 50 mM DTT/PBS was added and another 1 mL of protein collected.

Both supernatant and bound fraction proteins were then desalted by spinning the samples through 5 mL Zeba Spin Columns 7 k MWCO. Desalted samples were then lyophilized overnight and resuspended in NuPAGE loading buffer before being loaded onto NuPAGE gels (Invitrogen). Gels were run for 6 min at 180 V.

## Peptide identification by LC–MS/MS

Protein samples in gels were then prepared for and subjected to Label-free LC–MS by the Metabolomics and Proteomics Department in the Technology Facility at the University of York.

In-gel tryptic digestion was performed post-reduction with DTE and S-carbamidomethylation with iodoacetamide. Extracted peptides were loaded onto an mClass nanoflow UPLC system (Waters) equipped with a nanoEaze M/Z Symmetry 100 Å, C 18, 5 µm trap column (180 µm × 20 mm, Waters) and a PepMap, 2 µm, 100 Å, C 18, EasyNano nanocapillary column (75 mm × 500 mm, Thermo). The trap wash solvent was aqueous 0.05% (vol:vol) trifluoroacetic acid and the trapping flow rate was 15 µL/min. The trap was washed for 5 min before switching flow to the capillary column. Separation used gradient elution of two solvents: solvent A, aqueous 0.1% (vol:vol) formic acid; solvent B, acetonitrile containing 0.1% (vol:vol) formic acid. The flow rate for the capillary column was 300 nL/min and the column temperature was 40°C. The linear multi-step gradient profile was: 3%–10% B over 8 min, 10%–35% B over 115 min, 35%–99% B over 30 min, and then proceeded to wash with 99% solvent B for 4 min. The column was returned to initial conditions and re-equilibrated for 15 min before subsequent injections.

The nanoLC system was interfaced with an Orbitrap Fusion Tribrid mass spectrometer (Thermo) with an EasyNano ionization source (Thermo). Positive electrospray ionization mass spectrometry (ESI-MS) and MS 2 spectra were acquired using Xcalibur software (version 4.0, Thermo). Instrument source settings were: ion spray voltage, 1,900–2,100 V; sweep gas, 0 Arb; ion transfer tube temperature, 275°C. MS 1 spectra were acquired in the Orbitrap with 120,000 resolution, scan range: m/z 375–1,500; AGC target, 4e5; max fill time, 100 ms. The data-dependent acquisition was performed in top speed mode using a 1 s cycle, selecting the most intense precursors with charge states >1. Easy-IC was used for internal calibration. Dynamic exclusion was performed for 50 s post precursor selection and a minimum threshold for fragmentation was set at 5e3. MS 2 spectra were acquired in the linear ion trap with scan rate, turbo; quadrupole isolation, 1.6 m/z; activation type, HCD; activation energy: 32%; AGC target, 5e3; first mass, 110 m/z; max fill time, 100 ms. Acquisitions were arranged by Xcalibur to inject ions for all available parallelizable time.

Peak picking, database searching, and quantification of proteomic data from Thermo .raw files were performed using FragPipe (v19.1). Data were searched against a custom database of all coding regions of the *P. putredinis* NO1 genome appended with common contaminants and reversed sequences. The default LFQ-MBR workflow was used with the following modifications: precursor mass tolerance = ±3 ppm; fragment mass tolerance = 0.5 Da; IonQuant, feature detection m/z tolerance = 3 ppm; MBR RT tolerance = 7.5 min; add MaxLFQ, MBR FDR = 0.01, MBR min ions = 2. Final protein-level data were filtered to 1% FDR, a minimum protein probability of 0.99, and a minimum of two peptides. Molar percentage values were calculated for each protein in each sample as a percentage of the sum of MaxLFQ values for each sample.

## Quality control of proteomic data

Identified protein count, principal component analysis, and hierarchical clustering with a Canberra distance matrix and ward.D2 clustering was used to investigate proteomic data for all biological replicates for all substrates investigated. This analysis was performed in R studio 4.2.3 using the "ggplot2," "FactoMineR," "factoextra," "ggdendro," and "dendextend" packages (80–85). From this analysis, outliers were removed to provide a final data set for comparative analysis.

## Isolating the *P. putredinis* NO1 secretome

Proteins were filtered to produce the *P. putredinis* NO1 secretome using the "strict" filtering workflow developed previously (21). Secretome proteins were predicted to be extracellular by both BUSCA and DeepLoc localization prediction tools (20, 22), or were predicted to encode a secretion signal by SignalP, TargetP, and SecretomeP tools (23–25), and lacked more than one predicted transmembrane domain by TMHMM or contained a single predicted transmembrane helix with more than 10 amino acids of this helix occurring in the first 60 amino acids of the protein sequence (28).

## Comparing the *P. putredinis* NO1 secretome across substrates

All comparative analysis was performed in R studio v 4.2.3 (80), and analysis was repeated for the total proteome and the filtered secretome.

The number of proteins identified in at least one replicate across all substrates was compared between the bound and supernatant fraction using the "ggVennDiagram" package (86). To visualize differences across substrates, heatmaps were created to compare the average molar percentage of proteins across substrates using the "pheatmap" package (87). Principal component analysis was carried out to investigate replicate grouping within and between substrates using the "FactoMineR" and "factoextra" packages (82, 83). Canberra distance matrix calculation with the ward.D2 clustering was used to plot dendrograms to investigate replicate clustering within and between substrates using the "ggdendro", and "dendextend" packages (84, 85). GO

category and GO term annotation was performed using Blast2GO in the OmicsBox package (88, 89) and abundances were visualized and compared using the "ggplot2" package (81). CAZyme domain annotation was performed using the dbCAN server (43), and abundances were visualized and compared using the "ggplot2" package (81).

## Fluorescence-based activity-based protein profiling

Two microliters of triplex probe mixture (60 µM JJB376, CB644, SYF230) were added to 18 µL of secretome without added buffer (measured pH ~7.5). The reactions were incubated for 1 h at 30°C then quenched by the addition of 8 µL of 4× Laemmli buffer (Bio-Rad) and heating to 95°C for 2 min. Ten microliters of the resulting solution were separated over either a 10% SDS-PAGE gel or a 4%–20% gradient gel (Bio-Rad) at 200 V alongside PageRuler prestained protein ladder (Thermo Scientific). The resulting gels were transferred to a Typhoon 5 scanner without fixation and imaged using the Cy2, Cy3, and Cy5 laser/filter sets sequentially. The resulting images were integrated using ImageQuant (GE Healthcare).

## ACKNOWLEDGMENTS

The York Centre of Excellence in Mass Spectrometry was created thanks to a major capital investment through Science City York, supported by Yorkshire Forward with funds from the Northern Way Initiative, and subsequent support from ESPRC (EP/K039660/1; EP/M028127/1).

This work was funded by the Biotechnology and Biological Sciences Research Council (BBSRC), UK (BB/P027717/1, BB/W000695/1, BB/W003309/1, BB/S01196X/1). C.J.R.S. was supported by a CASE studentship from the BBSRC Doctoral Training Programme (BB/M011151/1) with Prozomix Ltd. G.J.D. is funded by the Royal Society Ken Murray Research Professor and G.J.D./H.S.O. though ERC-2020-SyG-951231 "Carbocentre."

C.J.R.S. conceptualized the investigation, analyzed the proteomic data, harvested culture supernatants for ABPP, and was a major contributor in writing the manuscript. N.G.S.M. performed the ABPP and imaged the gels, analyzed the ABPP results, and was a major contributor in writing the manuscript. D.R.L. aided the analysis and investigation and was a major contributor in writing the manuscript. N.C.O. helped to conceptualize the investigation and performed the proteomic experiment. A.A. aided in performing the proteomic experiment. A.S. aided in performing the proteomic experiment. A.D. performed the proteomic identification and helped in the analysis of the proteomic data. H.S.O. was a major contributor to the development of the ABPP methodology and contributed to writing. G.J.D. was a contributor to supervision and helped in writing the manuscript. N.C.B. was a major contributor to the conceptualization and supervision of the study in addition to making a major contribution to the writing of the manuscript.

## AUTHOR AFFILIATIONS

[1]Centre for Novel Agricultural Products, Department of Biology, University of York, York, United Kingdom
[2]York Structural Biology Laboratory, Department of Chemistry, The University of York, York, United Kingdom
[3]Bioscience Technology Facility, Department of Biology, University of York, York, United Kingdom
[4]Leiden Institute of Chemistry, Leiden University, Leiden, The Netherlands

## AUTHOR ORCIDs

Conor J. R. Scott  http://orcid.org/0000-0001-7404-7619
Daniel R. Leadbeater  http://orcid.org/0000-0002-2228-5604
Gideon J. Davies  http://orcid.org/0000-0002-7343-776X
Neil C. Bruce  http://orcid.org/0000-0003-0398-2997

## FUNDING

| Funder | Grant(s) | Author(s) |
|---|---|---|
| UKRI \| Biotechnology and Biological Sciences Research Council (BBSRC) | BB/P027717/1, BB/W000695/1, BB/W003309/1, BB/S01196X/1, BB/M011151/1 | Daniel R. Leadbeater Neil C. Bruce |
| Royal Society Ken Murray Research Professor | ERC-2020-SyG-951231 | Gideon J. Davies |
| UKRI \| Engineering and Physical Sciences Research Council (EPSRC) | EP/K039660/1, EP/M028127/1 | Adam Dowle |

## DATA AVAILABILITY

All proteomic data generated during this research are available at MassIVE MSV000092129.

## ADDITIONAL FILES

The following material is available online.

### Supplemental Material

**Figure S1 (Spectrum03943-23-s0001.tiff).** Investigating the distribution of protein.
**Figure S2 (Spectrum03943-23-s0002.tiff).** Variation in the *P. putredinis* NO1 secretome.
**Figure S3 (Spectrum03943-23-s0003.tiff).** Clustering of replicates of the *P. putredinis* NO1 secretome.
**Figure S4 (Spectrum03943-23-s0004.tiff).** Differences in *P. putredinis* NO1 glycoside hydrolase production.

### Open Peer Review

**PEER REVIEW HISTORY (review-history.pdf).** An accounting of the reviewer comments and feedback.

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
