## [Reviewer comments · Microbiology Spectrum]

Microbiology Spectrum

***Parascedosporium putredinis* NO1 tailors its secretome for different lignocellulosic substrates**

Conor Scott, Nicholas McGregor, Daniel Leadbeater, Nicola Oates, Janina Hoßbach, Amira Abood, Alexander Setchfield, Adam Dowle, Herman Overkleeft, Gideon Davies, and Neil Bruce

Corresponding Author(s): Conor Scott, University of York

Review Timeline:

Submission Date:	November 14, 2023
Editorial Decision:	March 25, 2024
Revision Received:	April 9, 2024
Accepted:	April 19, 2024

Editor: Sudhir Singh

Reviewer(s): Disclosure of reviewer identity is with reference to reviewer comments included in decision letter(s). The following individuals involved in review of your submission have agreed to reveal their identity: Vinod Kumar (Reviewer #3); Anmoldeep Randhawa (Reviewer #4)

Transaction Report:

DOI: <https://doi.org/10.1128/spectrum.03943-23>

Re: Spectrum03943-23 (*Parascedosporium putredinis* NO1 tailors its secretome for different lignocellulosic substrates)

Dear Mr. Conor James Richard Scott:

Thank you for the privilege of reviewing your work. Below you will find my comments, instructions from the Spectrum editorial office, and the reviewer comments.

Revision Guidelines

Sincerely,
Sudhir Singh
Editor
Microbiology Spectrum

Reviewer #2 (Comments for the Author):

1. Why authors did specifically select *Parascedosporium putredinis* NO1 for the current study? Please justify.
2. How did the authors characterize and identify the isolate as *Parascedosporium putredinis*. Please provide the full details in material and methods.
3. A statistical analysis section is missing from the material and methods. Please update.

4. Authors should include a separate section describing the limitations of the current study.
5. Please include the statistics such as p values and control details in figure legends, wherever applicable.

Reviewer #3 (Comments for the Author):

The article titled "Parascedosporium putredinis NO1 tailors its secretome for different lignocellulosic substrates" is a commendable piece of research that underscores the significance of the fungus *Parascedosporium putredinis* NO1. The authors delved into the examination of the fungus's secretome, presenting it as a promising candidate for identifying enzymes conducive to breaking down challenging lignocellulosic feedstocks. The study aptly emphasizes the influence of various substrate specificities on the secretome of the examined fungus. While this research shows promise and merits acceptance after peer review, there are several essential queries that need addressing before formal acceptance.

1. The total number of proteins extracted from all the substrates has been mentioned and represented in Venn diagram. However, proteins across bound and supernatant fractions among particular substrate has not been mentioned. Please mention the number.
2. What was the purpose of taking bound and supernatant proteins. It may be like when substrate is added in the medium, the fungi try to secrete enzymes or proteins in the culture media in order to degrade/hydrolyse the substrate by binding the substrate and there will be continuity of substrate bound and free enzymes in the media. Thus, this may represent that the proteins remain actually the same. However, their secretion and expression may differ and will be influenced.
3. In Line No. 26 add reference.
4. Line No. 212. The differences between the substrates (may be structural and physical) were not highlighted or mentioned which may have influence on the studied secretome. This may get further analysed by principle component analysis by including physical and structural parameters of the substrates as variables and to measure the contribution of these parameters (using contrib option in PCA) influencing secretome.
5. In Line 228. Why is there a variation in bound and supernatant proteins? It may be addressed like different substrates activate the different genes with different intensity to synthesize different proteins with different abundance. And this is not dependent on the bound and supernatant fractions rather depend on structure and composition of substrates (with different concentrations of lignin, polysaccharides etc.).
6. In line 231. The assumption or reason made in this line indicates that the structure plays an important role in activating the transcriptome of fungi.
7. In Line 236. Does CAZymes contributed the "most" or the "least" here? It is contradictory to figure 3. Please correct. It can be seen in the figure 3 that supernatant fraction contributes most for kraft lignin.
8. In Line 260-261. This is results are again contrasting with the figure 4. Please correct
9. In Line 221. Discussion is poor in this section, needs to be enhanced with more references.
10. Sentences at line no. 372 and line no. 360 are contradictory. Please discuss.
11. In the Line 380. Why wasn't this approach employed on substrate bound proteins? Provide explanation.
12. In Line 414. This is in line and reviewer expects the similar explanation as with the comment 5.
13. In Figure 8, RS Fig is poor, and also ladder has not been named in any subfigures in Fig8.
14. Fig 1, Fig5, Fig9, and Supplementary Fig 2, 3 are missing.

Reviewer #4 (Comments for the Author):

Conor JR Scott and colleagues performed detailed proteomic analysis of soluble and insoluble culture fractions of *P. putredinis* NO1 grown on six lignocellulosic substrates, and found variation in CAZYmes abundance under these conditions. They verified their results by ABPP approach. The proteomic analysis is well conducted The manuscript is long but well written. The minor comments are :

1. Since *P. putredinis* NO1 was grown in the presence of six different substrates and variations were found in CAZyme proportions, the authors should suggest the cocktail of major enzymes that might be useful for saacharification of the six substrates. Moreover, how the change in Cazyme proportion helps in saccharification of the give substrate.
2. The authors should add the number of samples used to carry out proteomic analysis in figure captions along with cut-offs for p-value and fold change.

The article titled "*Parascedosporium putredinis* NO1 tailors its secretome for different lignocellulosic substrates" is a commendable piece of research that underscores the significance of the fungus *Parascedosporium putredinis* NO1. The authors delved into the examination of the fungus's secretome, presenting it as a promising candidate for identifying enzymes conducive to breaking down challenging lignocellulosic feedstocks. The study aptly emphasizes the influence of various substrate specificities on the secretome of the examined fungus. While this research shows promise and merits acceptance after peer review, there are several essential queries that need addressing before formal acceptance.

1. The total number of proteins extracted from all the substrates has been mentioned and represented in Venn diagram. However, proteins across bound and supernatant fractions among particular substrate has not been mentioned. Please mention the number.
2. What was the purpose of taking bound and supernatant proteins. It may be like when substrate is added in the medium, the fungi try to secrete enzymes or proteins in the culture media in order to degrade/hydrolyse the substrate by binding the substrate and there will be continuity of substrate bound and free enzymes in the media. Thus, this may represent that the proteins remain actually the same. However, their secretion and expression may differ and will be influenced.
3. In Line No. 26 add reference.
4. Line No. 212. The differences between the substrates (may be structural and physical) were not highlighted or mentioned which may have influence on the studied secretome. This may get further analysed by principle component analysis by including physical and structural parameters of the substrates as variables and to measure the contribution of these parameters (using contrib option in PCA) influencing secretome.
5. In Line 228. Why is there a variation in bound and supernatant proteins? It may be addressed like different substrates activate the different genes with different intensity to synthesize different proteins with different abundance. And this is not dependent on the bound and supernatant fractions rather depend on structure and composition of substrates (with different concentrations of lignin, polysaccharides etc.).
6. In line 231. The assumption or reason made in this line indicates that the structure plays an important role in activating the transcriptome of fungi.
7. In Line 236. Does CAZymes contributed the "most" or the "least" here? It is contradictory to figure 3. Please correct. It can be seen in the figure 3 that supernatant fraction contributes most for kraft lignin.
8. In Line 260-261. This is results are again contrasting with the figure 4. Please correct
9. In Line 221. Discussion is poor in this section, needs to be enhanced with more references.
10. Sentences at line no. 372 and line no. 360 are contradictory. Please discuss.
11. In the Line 380. Why wasn't this approach employed on substrate bound proteins? Provide explanation.
12. In Line 414. This is in line and reviewer expects the similar explanation as with the comment 5.
13. In Figure 8, RS Fig is poor, and also ladder has not been named in any subfigures in Fig8.
14. Fig 1, Fig5, Fig9, and Supplementary Fig 2, 3 are missing.

Page and line numbers referenced in this response correspond to the “Marked-Up
Manuscript” document.
**Reviewer #2**
**Comment:** Why authors did specifically select *Parascedosporium putredinis* NO1 for the
current study? Please justify.
*Response: Parascedosporium putredinis* NO1 was previously identified from a mixed
microbial community grown on wheat straw, where the ascomycete dominated the fungal
population in the later stages of the community. This suggested that *P. putredinis* NO1 could
utilise the more recalcitrant components of lignocellulose and the secretome of this fungus
could contain new enzymes involved in degrading plant biomass. Indeed, a new oxidase
enzyme cleaving the major β -ether units in lignin was identified from this fungus. Therefore,
*P. putredinis* NO1 was selected to understand the lignocellulose-degrading secretome of a
fungus which may target the most difficult to degrade components of lignocellulose while
utilising new enzymes to do so. These concepts are included in the introduction:
Page 6, line 123: “*Parascedosporium putredinis* NO1 is a soft-rot ascomycete identified from
a mixed microbial community grown on wheat straw (17). *P. putredinis* NO1 dominated the
fungal population in the later stages of the community culture, suggesting an ability to
metabolise the more recalcitrant carbon sources in the substrate. This makes *P. putredinis*
NO1 an interesting candidate for the identification of new lignocellulose degrading enzymes.
Indeed, a new oxidase enzyme was recently identified in the *P. putredinis* NO1 secretome
which cleaves the major β -ether units in lignin to enhance the digestibility of wheat straw
releasing triclin, coumaric acid, and vanillic acid in the process (17).”
*However, an addition has been made in the introduction to clarify why this is industrially*
*relevant:*
Page 6, line 130: “Understanding the secretome response of a fungus which produces new
lignocellulose-degrading enzymes, and which may target the recalcitrant components of
lignocellulose, will be important for the development of effective methods to valorise
lignocellulose.”
**Comment:** How did the authors characterize and identify the isolate as *Parascedosporium*
*putredinis*. Please provide the full details in material and methods.
*Response: P. putredinis* NO1 was identified, isolated, and characterised previously in a
publication referenced in the materials and methods section (Oates et al. 2021). The “Strain
isolation” section of the Experimental methods has now been extended to describe this
process:
Page 23, line 556: “A mixed microbial community grown on wheat straw was characterised
by 16S ribosomal and ITS region amplicon sequencing over 8 weeks of growth. ITS reads
assigned to the genus *Graphium* dominated the eukaryotic community after 4 weeks of
incubation. *P. putredinis* NO1, a synamorph of *Graphium*, was readily cultivated from this
community on potato dextrose agar plates.”
**Comment:** A statistical analysis section is missing from the material and methods. Please
update.
*Response: The analysis performed in this paper was primarily to summarise the profiles of*
*enzyme activities present in the P. putredinis NO1 secretome and to compare these across*
*substrates. No statistical tests were performed to compare GO category assignment or*
*CAZyme class assignment proportions across substrates, as these annotations were used to*
*profile the secretome activities on each substrate and visualise how they differ between*
*substrates.*
*In the manuscript, it is admitted that the major limitation of the proteomic comparison at a*
*single timepoint is that differences between secretomes could be attributed to differences in*
*fungal growth stage and this is why the fluorescent labelling technique was used to*
*demonstrate secretome differences over time. With this limitation considered when*
*investigating and visualising the secretome profiles, statistical analysis seems unnecessary.*
**Figures 3 and 6** *have however been reconsidered as the information presented by both bar*
*charts and pie charts seemed redundant. The bar charts have now been removed so that*
*GO category proportions in **Figure 3** and CAZyme class proportions in **Figure 6** are now*
*presented in pie chart format for the bound and supernatant fraction separately. This makes*
*the figures much clearer and reduces confusion about statistical analysis. The figure legends*
*have been updated to reflect this:*
Page 29, line 723: **Figure 3. Differences in functional categories of the *P. putredinis***
**NO1 secretome across lignocellulosic substrates.** The molar percentage abundance of
proteins identified as CAZymes, that were not identified as CAZymes but assigned to GO
categories, or that were not annotated as CAZymes or with GO categories (Unknown) were
calculated proportionally for each substrate of the bound (A) and supernatant fractions (B) of
the *P. putredinis* NO1 secretome.”
Page 30, line 752: “**Figure 6. Differences in proportional catalytic CAZyme class**
**abundance of the *P. putredinis* NO1 secretome across lignocellulosic substrates.** The
molar percentage abundance of proteins belonging to each catalytic class of CAZyme were
calculated proportionally to the total abundance of CAZymes for each substrate for the
bound (A) and supernatant fractions (B) of the *P. putredinis* NO1 secretome.”
*In Figures 4 and 5, the molar percentage contributions of the top ten GO term assignments*
*are presented for each substrate for the bound (Figure 4) and supernatant (Figure 5)*
*fractions separately. As the GO terms appearing in the top ten for each substrate varies,*
*statistical analysis is not applicable here. Standard error bars on the bar charts in these*
*figures are still useful to provide readers with an idea of the variation between replicates of*
*secretomes from the same substrate and fraction. The number of replicates has been*
*clarified in the figure legends:*
Page 30, line 737: “**Figure 4. Proportional abundances of enzyme activities in the**
**bound fraction of the *P. putredinis* NO1 secretome.** The molar percentage abundance of
proteins assigned to GO terms were calculated proportionally for the bound fraction of the *P.*
*putredinis* NO1 secretome after 4 days of growth on 6 lignocellulosic substrates. EF = Empty
fruit bunch (A), LI = Kraft lignin (B), RS = Rice straw (C), SC = Sugar cane bagasse (D), WB
= Wheat bran (E), WS = Wheat straw (F). Average molar percentage \pm standard error (n = 3,
n = 2 for SC substrate).”
Page 30, line 744: “**Figure 5. Proportional abundances of enzyme activities in the**
**supernatant fraction of the *P. putredinis* NO1 secretome.** The molar percentage
abundance of proteins assigned to GO terms were calculated proportionally for the
supernatant fraction of the *P. putredinis* NO1 secretome after 4 days of growth on 6
lignocellulosic substrates. EF = Empty fruit bunch (A), LI = Kraft lignin (B), RS = Rice straw
(C), SC = Sugar cane bagasse (D), WB = Wheat bran (E), WS = Wheat straw (F). Average
molar percentage \pm standard error (n = 3, n = 2 for SC, LI, and WB substrates).”
**Comment:** Authors should include a separate section describing the limitations of the
current study.
*Response: Although it is felt that a dedicated section for the limitations of the study is*
*unnecessary, the limitations of some aspects of the study have been clarified with the*
*following additions in the Results and discussion sections:*
Page 9, line 199: “The limitations of this workflow have been addressed previously (21), and
it is accepted that some intracellular proteins may remain in the predicted secretome.
However, the increased proportion of proteins annotated with extracellular functions
achieved through this workflow demonstrates the removal of large numbers of intracellular
proteins and therefore improves any interpretation of the *P. putredinis* NO1 secretome.”
Page 10, line 216: “The lower number of replicates for the supernatant fractions of some
substrates, and the lower protein counts in the supernatant fractions of some replicates
reflects the difficulty in applying a single method of proteome harvest to cultures containing
different substrates and therefore different conditions. This could also limit the ability to
perform statistical comparisons between substrates. However, for the purpose of this study
these replicates were still suitable for the visualisation and comparison of the overall profiles
of the *P. putredinis* NO1 secretome across substrates”
Page 12, line 274: “The persistence of proteins with predicted intracellular functions was
reported previously in the development of the secretome isolation workflow applied here
(21). The interference of these potentially intracellular proteins could have been reduced
through stricter filtering when obtaining the *in silico* secretome, however this could risk
removing unknown proteins that are important for lignocellulose degradation”
Page 17, line 418: “One limitation of the current study was that it was possible for the
observed differences in secretome profiles to be a product of different fungal growth stage
on each substrate, as all data was harvested at a single timepoint. An alternative approach
was therefore taken to confirm that differences in the *P. putredinis* NO1 secretome are
maintained over time.”
Page 21, line 510: “Considering the potential lignin-degrading lifestyle of *P. putredinis* NO1,
from which a new lignin-degrading oxidase has been identified previously (17), the
investigation of oxidative enzymes in the secretome and their variation dependent on
substrate and over time would be of great interest. However, at present the use of ABPP
techniques to probe fungal secretomes as was performed here is limited to hydrolytic
enzymes.”
**Comment:** Please include the statistics such as p values and control details in figure
legends, wherever applicable.
*Response:* As explained above, statistical analysis was not used when profiling and
visualising the *P. putredinis* NO1 secretome and therefore p values have not been included
in the figure legends. Regarding the setup of the proteomic investigation, the secretomes
*were harvested, profiled, and visualised for both bound and supernatant fractions when P.*
*putredinis NO1 was grown on the six lignocellulosic substrates presented in this manuscript*
*and no controls conditions were investigated.*
**Reviewer #3**
**Comment:** The total number of proteins extracted from all the substrates has been
mentioned and represented in Venn diagram. However, proteins across bound and
supernatant fractions among particular substrate has not been mentioned. Please mention
the number.
*Response: A passage stating the number of proteins identified in the bound fraction,*
*supernatant fraction, and shared between both for each substrate individually:*
Page 9, line 188: "Considering each substrate individually; for empty fruit bunch 107 proteins
were identified in the bound fraction, 109 exclusively in the supernatant fraction, and 58
shared between both; for Kraft lignin 85 proteins were identified in the bound fraction, 20 in
the supernatant fraction, and 14 shared between both; for rice straw 130 proteins were
identified in the bound fraction, 100 in the supernatant fraction, and 60 shared between both;
for sugar cane bagasse 105 proteins were identified in the bound fraction, 104 in the
supernatant fraction, and 66 shared between both; for wheat bran 72 proteins were identified
in the bound fraction, 150 in the supernatant fraction, and 47 shared between both; wheat
straw 101 proteins were identified in the bound fraction, 144 in the supernatant fraction, and
77 shared between both."
**Comment:** What was the purpose of taking bound and supernatant proteins. It may be like
when substrate is added in the medium, the fungi try to secrete enzymes or proteins in the
culture media in order to degrade/hydrolyse the substrate by binding the substrate and there
will be continuity of substrate bound and free enzymes in the media. Thus, this may
represent that the proteins remain actually the same. However, their secretion and
expression may differ and will be influenced.
*Response: Both bound and supernatant fractions were harvested to attempt to capture the*
*lignocellulose-degrading secretome of P. putredinis NO1 as fully as possible. Many fungal*
*lignocellulose-degrading proteins contain binding domains (CBMs) which associate the*
*proteins tightly with the substrate. If only the supernatant fraction was harvested, then many*
*of these proteins may have been missed. The differences in the protein complements*
*identified in each fraction suggests that this strategy did effectively identify more proteins*
*than would have been identified by only taking one fraction. This reasoning has been*
*clarified when discussing CBMs:*
Page 16, line 376: "Harvesting the bound fraction of lignocellulose-degrading secretomes
was performed in an attempt to capture proteins with such binding capacity, which may have
been missed by harvesting proteins from the soluble supernatant fraction only."
**Comment:** In Line No. 26 add reference.
*Response: This line corresponds to the data availability section referencing the deposition of*
*the raw proteomic data. The data is not currently publicly available, but access can be given*
*to reviewers upon request and the data can be made public before publication. The code*
*MSV000092129 is the reference number for this data set.*
**Comment:** Line No. 212. The differences between the substrates (may be structural and
physical) were not highlighted or mentioned which may have influence on the studied
secretome. This may get further analysed by principle component analysis by including
physical and structural parameters of the substrates as variables and to measure the
contribution of these parameters (using contrib option in PCA) influencing secretome.
*Response: Earlier in the passage the similar proportions of cellulose, hemicellulose, and*
*lignin across many of the substrates is referenced:*
Page 10, line 223: "This is especially interesting considering the somewhat similar
proportions of cellulose, hemicellulose, and lignin for the empty fruit bunch from oil palm
(29), rice straw (30), sugar cane bagasse (31), and wheat straw (32)."
*When discussing wheat bran and kraft lignin substrates, the distinct compositional*
*differences of these substrates is also referenced multiple times:*
Page 11, line 236: "Wheat bran has a different overall composition compared to the other
lignocellulosic substrates with an increased starch content, correspondingly decreased
proportions of cellulose and hemicellulose, and a reduced lignin content (36). The kraft lignin
substrate is expected to contain minimal polysaccharides and instead is composed
predominantly of branched polymeric lignin containing native lignin bonding patterns and
new structures created during the kraft pulping process (37)."
Page 13, line 298: "Once again, kraft lignin and wheat bran, the two most compositionally
distinct substrates explored here demonstrate the most distinct secretome activity profiles."
Page 14, line 330: “Crystalline cellulose is present in lower levels or is absent from wheat
bran and kraft lignin substrates, and this may reflect a reduced production of AA class
CAZymes involved in lytic polysaccharide monooxygenase (LPMO) systems which degrade
crystalline cellulose oxidatively (36, 37, 39, 40).”
Page 16, line 382: “Indeed some expected patterns can be seen in the bound fraction, such
as the relatively lower abundance of CBM1 proteins, involved in crystalline cellulose binding,
to kraft lignin and wheat bran substrates which have reduced proportions of cellulose in
comparison to the other substrates investigated (36, 37, 52)”
*It is accepted that subtle structural and physical differences outside of the overall*
*composition of the substrates likely influenced the differences in the P. putredinis NO1*
*secretome. However an analysis of the influence of these features seems outside the scope*
*of this work, which is to instead present the highly variable and potentially tailored*
*lignocellulose-degrading secretome of P. putredinis NO1. This has now been addressed in*
*the manuscript:*
Page 11, line 234: “Further study of these more subtle structural, compositional, and physical
differences could improve the understanding of how fungal secretomes are influenced and
tailored accordingly.”
**Comment:** In Line 228. Why is there a variation in bound and supernatant proteins? It may
be addressed like different substrates activate the different genes with different intensity to
synthesize different proteins with different abundance. And this is not dependent on the
bound and supernatant fractions rather depend on structure and composition of substrates
(with different concentrations of lignin, polysaccharides etc.).
*Response: It is expected, as discussed above, that the compositions and structural and*
*physical differences of the substrates influenced the proteins present in the P. putredinis*
*NO1 secretome. It is also expected that these differences between substrates occur for both*
*the bound and supernatant fractions of the secretome. In this work multiple substrates have*
*been investigated and the focus of the manuscript is on the differences between these*
*substrates and not between the bound and supernatant fractions for individual substrates.*
*However, the differences between the bound and supernatant fractions are clear when*
*looking at individual substrates in all the proteomic data presented, once again validating the*
*approach of capturing both fractions which contain their own unique complements of*
*lignocellulose-degrading proteins. This concept has been reinforced in the text:*
Page 11, line 252: "The differences between bound and supernatant fractions for individual
substrates validates further the importance of capturing both fractions when exploring a
secretome in this context, as each fraction clearly contains distinct profiles of enzyme
activities."
**Comment:** In line 231. The assumption or reason made in this line indicates that the
structure plays an important role in activating the transcriptome of fungi.
*Response: It is agreed that physical structure likely influences the content of the P. putredinis*
*NO1 secretome and this has been clarified in the text:*
Page 12, line 262: "Structural aspects of lignocellulose such as cellulose structure and
accessibility, pore size and distribution, the extent and nature of lignin-carbohydrate
complexes have all been reported to influence the efficiency of enzymatic hydrolysis (38).
However, the relationship between such physical characteristics and microbial enzyme
production is less well understood."
**Comment:** In Line 236. Does CAZymes contributed the "most" or the "least" here? It is
contradictory to figure 3. Please correct. It can be seen in the figure 3 that supernatant
fraction contributes most for kraft lignin.
*Response: This was a mistake and should have referred to the supernatant fraction for Kraft*
*lignin. This has now been amended:*
Page 12, line 266: "Surprisingly, CAZymes contributed the most to total abundance in the
supernatant fraction for kraft lignin"
**Comment:** In Line 260-261. This is results are again contrasting with the figure 4. Please
correct
*Response: It seems that the results here are not contrasting, the following statement about*
*membrane GO annotations for the bound fraction of Kraft lignin and wheat bran substrates is*
*true:*
Page 13, line 295: "In the bound fraction, membrane GO annotations contributed the highest
abundance of any GO term for kraft lignin (**Figure 4B**) and wheat bran (**Figure 4E**), which
may reflect the reduced CAZyme proportions seen previously for these substrates in this
fraction."
*It has been clarified in the next sentence that the secretomes being referenced do not*
*include kraft lignin:*
Page 13, line 299: “Aside from cell surface proteins, abundant intracellular annotations were
largely absent from most of the secretomes except for the kraft lignin substrate.”
**Comment:** In Line 221. Discussion is poor in this section, needs to be enhanced with more
references.
*Response: Substantial expansion of the discussion has now been included in this section:*
Page 12, line 262: “Structural aspects of lignocellulose such as cellulose structure and
accessibility, pore size and distribution, the extent and nature of lignin-carbohydrate
complexes have all been reported to influence the efficiency of enzymatic hydrolysis (38).
However, the relationship between such physical characteristics and microbial enzyme
production is less well understood.”
Page 12, line 266: “Surprisingly, CAZymes contributed the most to total abundance in the
supernatant fraction for kraft lignin which, although is poorly soluble in water, lacks the
polysaccharide chains targeted by binding modules (33, 39).”
Page 12, line 284: “For all substrates in both the bound (**Figure 4**) and supernatant (**Figure**
**5**) fractions, the top GO term annotations contributing to total abundance were
predominantly extracellular activities associated with hydrolytic breakdown of lignocellulose,
agreeing with the polysaccharide abundance in many of the substrates (29-32, 36)”
Page 13, line 288: “In both fractions, many substrates also included membrane annotations
in the top ten most abundant GO terms. This may result from the identification of proteins in
the workflow which may be targeted to the secretory pathway, but that are not necessarily
secreted and instead fulfil roles in membranes (40). Another possibility is the binding and
extraction of extracellular cell surface proteins by the biotin labelling technique, as
biotinylation is a common technique for surface proteomic studies (41).”
Page 13, line 311: “Finally, the GO annotation profile of the bound fraction for wheat bran
(**Figure 4E**) distinctly contained chitin binding and catabolising annotations. Wheat bran has
been investigated for its use as a substrate for the induction of chitinase production in fungal
species previously and perhaps a similar induction is occurring for *P. putredinis* NO1 (42).”
**Comment:** Sentences at line no. 372 and line no. 360 are contradictory. Please discuss.
*Response: Earlier in the passage, the suggestion is made that the presence of*
*polysaccharide degrading enzymes in the kraft lignin secretome may represent a general*
*production of these enzymes. As it has been demonstrated previously for Trichoderma*
*reesei that spore germination can lead to CAZyme production before a substrate has been*
*detected.*
*However, this idea does not contradict the suggestion of a tailored secretome as it is*
*possible that the action of enzymes produced irrespective of substrate may induce further*
*CAZyme production which leads to the tailoring of the secretome.*
*This is discussed later in the ABPP section, but has now been clarified earlier:*
Page 17, line 402: "Indeed, it has been observed in *T. reesei* that spore germination alone
can lead to substantial upregulation of polysaccharide degrading CAZymes to prepare the
fungus for a habitat containing plant biomass (62). The GH10, GH26, GH55, and CE15
families produced in relatively high abundance for kraft lignin possibly represent a similar
response. It is possible that the action of these enzymes, which may be produced
irrespective of substrate, release oligo- or mono-saccharides that could lead to induction and
tailoring of the secretome (35, 51, 61)."
**Comment:** In the Line 380. Why wasn't this approach employed on substrate bound
proteins? Provide explanation.
*Response: An explanation has now been included in the text*
Page 18, line 437: "The biotin-labelling approach used in the proteomic experiments to
sample bound fractions involves protein denaturation and is therefore incompatible with this
ABPP technique which requires active site binding."
**Comment:** In Line 414. This is in line and reviewer expects the similar explanation as with
the comment 5.
*Response: It has been clarified in the text that the distinct fluorescent profile for rice straw*
*samples could be related to the unique structural properties of this substrate:*
Page 19, line 463: "Perhaps the distinct fluorescent profile observed here for rice straw may
result from the unique structural properties of rice straw, such as its high silica content which
may prevent access to polysaccharides (71)."
**Comment:** In Figure 8, RS Fig is poor, and also ladder has not been named in any
subfigures in Fig8.
*Response: The image quality is not lower for the rice straw sample, instead there is an*
*absence of hydrolytic enzymes in these samples and therefore a lack of fluorescent bands.*
*This distinct lack of hydrolytic enzymes, at least those targeted by the probes used here, is*
*discussed in the text:*
Page 19, line 452: “For rice straw, this production remained low for the entire time course
with low levels of xylanase and β -glucosidase and only a single cellulase detected at low
abundances. Although the production profiles of these GHs was demonstrated to vary by
species, it is worth noting that transcriptomic analysis of another ascomycete, *Thielavia*
*terrestris*, only found a single cellulase gene to be in the most highly expressed genes when
grown on rice straw compared to eleven cellulases for growth on Avicel (68, 70). It was also
observed that compared to growth on glucose, predominantly oxidative AA LPMO family
CAZymes and hemicellulose-active enzymes were up-regulated during *T. terrestris* growth
on rice straw (70). The reason for the lack of cellulose targeting GHs also observed here is
unclear especially considering the reported induction of cellulases from both cellobiose and
xylose in other ascomycetes (35, 51, 61), but may suggest a predominantly oxidative
approach to rice straw deconstruction instead. Perhaps the distinct fluorescent profile
observed here for rice straw may result from the unique structural properties of rice straw,
such as its high silica content which may prevent access to polysaccharides (71). Relatively
high abundances of AA3 and AA12 family CAZymes were observed in the rice straw
secretome supernatant fraction, where the probes are deployed (**Figure 7**). These oxidative
enzymes act within LPMO systems, however relatively low abundances of the AA9 LPMOs
were also observed. Further investigation specifically into growth on rice straw would be
required to fully understand the strategy of lignocellulose breakdown adopted here by *P.*
*putredinis* NO1, but is warranted considering the abundance and environmental issues
posed by the vast amounts of rice straw generated annually and burnt on the field (72, 73).”
*The ladder used has now been added to the figure legend:*
Page 31, line 771: “**Figure 8. Differences in *P. putredinis* NO1 glycoside hydrolase**
**production over time visualised with activity-based probes.** Fluorescence imaging
following SDS-PAGE is shown for single replicates of samples of culture supernatants taken
at days 3, 4, 5, 6, 7, and 10 of growth were treated with a triplex probe mixture targeting
cellulases, xylanases, and retaining β -glucosidases. EF = Empty fruit bunch, RS = Rice
straw, SC = Sugar cane bagasse, WB = Wheat bran, WS = Wheat straw, alongside
PageRuler prestained protein ladder. Secretome were stained in triplicate, a single replicate
is shown here.”
*And the ladder used is also mentioned now in the Methods:*
Page 27, line 676: “The reactions were incubated for 1 hour at 30 °C then quenched by the
addition of 8 µL of 4x Laemmli buffer (Bio-Rad) and heating to 95 °C for 2 minutes. 10 µL of
the resulting solution was separated over either a 10 % SDS-PAGE gel or a 4-20 % gradient
gel (Bio-Rad) at 200 V alongside PageRuler prestained protein ladder (Thermo Scientific).”
**Comment:** Fig 1, Fig5, Fig9, and Supplementary Fig 2, 3 are missing.
*Response: All figures were included in the submitted merged manuscript file, all*
*supplementary figures were also attached.*
**Reviewer #4**
**Comment:** Since *P. putredinis* NO1 was grown in the presence of six different substrates
and variations were found in CAZyme proportions, the authors should suggest the cocktail of
major enzymes that might be useful for saacharification of the six substrates. Moreover, how
the change in Cazyme proportion helps in saccharification of the give substrate.
*Response: Demonstrations of the variation in the activity profiles of the lignocellulose-*
*degrading secretome of P. putredinis NO1 are made throughout the paper. The broader*
*impact of such work is to improve the understanding of fungal lignocellulose conversion*
*which could aid the development of efficient processes for the valorisation of plant biomass.*
*However, it is outside the scope of the current manuscript to suggest designs six enzymatic*
*cocktails for each substrate. In fact, the analysis has revealed the complexity of the fungal*
*response to different susbtrates, even when compositionally similar. This highlights the*
*weakness of applying generic enzyme cocktails to different substrates, and the difficulty in*
*disentangling the optimal solutions for individual substrates. Through investigations like*
*those carried out in this manuscript, progress towards tailored commercial cocktails can*
*begin. This reasoning has been reinforced in the conclusion:*
Page Number 22, line 545: “The complexity of fungal lignocellulose-degrading responses
revealed here highlights the shortcoming of generic commercial cocktails for application to
different substrates, but also the difficulty in disentangling enzyme effects and designing
optimal tailored cocktails.”
**Comment:** The authors should add the number of samples used to carry out proteomic
analysis in figure captions along with cut-offs for p-value and fold change.
*Response: As explained in the previous response to Reviewer #2, statistical analysis was*
*not performed when visualising the secretome profiles. However, the replicate numbers have*
*been added to figure legends:*
Page 30, line 737: “**Figure 4. Proportional abundances of enzyme activities in the**
**bound fraction of the *P. putredinis* NO1 secretome.** The molar percentage abundance of
proteins assigned to GO terms were calculated proportionally for the bound fraction of the *P.*
*putredinis* NO1 secretome after 4 days of growth on 6 lignocellulosic substrates. EF = Empty
fruit bunch (**A**), LI = Kraft lignin (**B**), RS = Rice straw (**C**), SC = Sugar cane bagasse (**D**), WB
= Wheat bran (**E**), WS = Wheat straw (**F**). Average molar percentage \pm standard error (n = 3,
n = 2 for SC substrate).”
Page 30, line 744: “**Figure 5. Proportional abundances of enzyme activities in the**
**supernatant fraction of the *P. putredinis* NO1 secretome.** The molar percentage
abundance of proteins assigned to GO terms were calculated proportionally for the
supernatant fraction of the *P. putredinis* NO1 secretome after 4 days of growth on 6
lignocellulosic substrates. EF = Empty fruit bunch (**A**), LI = Kraft lignin (**B**), RS = Rice straw
(**C**), SC = Sugar cane bagasse (**D**), WB = Wheat bran (**E**), WS = Wheat straw (**F**). Average
molar percentage \pm standard error (n = 3, n = 2 for SC, LI, and WB substrates).”

Re: Spectrum03943-23R1 (*Parascedosporium putredinis* NO1 tailors its secretome for different lignocellulosic substrates)

Dear Mr. Conor JR Scott:

Your manuscript has been accepted, and I am forwarding it to the ASM production staff for publication. Your paper will first be checked to make sure all elements meet the technical requirements. ASM staff will contact you if anything needs to be revised before copyediting and production can begin. Otherwise, you will be notified when your proofs are ready to be viewed.

Sincerely,
Sudhir Singh
Editor
Microbiology Spectrum

Reviewer #2 (Public repository details (Required)):

Secretome data should be submitted.

Reviewer #4 (Comments for the Author):

Response to reviewer's comments was found satisfactory